



# Marine CO₂ system variability along the Inside Passage of the Pacific Northwest coast of North America determined from an Alaskan ferry

Wiley Evans[1,*], Geoffrey T. Lebon[2,3], Christen D. Harrington[4], Yuichiro Takeshita[5], Allison Bidlack[6]

5   [1]Hakai Institute, Heriot Bay, BC, V0P 1H0, Canada
[2]Pacific Marine Environmental Laboratory, National Oceanic and Atmospheric Administration, Seattle, 98115, USA
[3]Cooperative Institute for Climate, Ocean, & Ecosystem Studies, University of Washington, 98195, USA
[4]Alaska Marine Highway, Department of Transportation, Ketchikan, AK, 99901, USA
[5]Monterey Bay Aquarium Research Institute, Moss Landing, 95039, USA
10   [6]Alaska Coastal Rainforest Center, University of Alaska Southeast, Juneau, AK, 99801, USA

*Correspondence to*: Wiley Evans (wiley.evans@hakai.org)





**Abstract** Information on marine $CO_2$ system variability has been limited along the Inside Passage of the Pacific Northwest coast of North America despite the region's rich biodiversity, abundant fisheries, and developing aquaculture industry. Beginning in 2017, the Alaska Marine Highway System M/V *Columbia* has served as a platform for surface underway data collection while conducting twice weekly ~1600-km transits between Bellingham, Washington and Skagway, Alaska. This dataset allowed for the assessment of marine $CO_2$ system patterns along the Inside Passage, including quantification of the relative importance of key drivers in shaping $pCO_2$ variability. Surface water pH and aragonite saturation state ($\Omega_{arag}$) were determined using the $pCO_2$ data with alkalinity from a regional salinity-based relationship, which was evaluated with discrete seawater samples and underway pH measurements. Low pH and corrosive ($\Omega_{arag} < 1$) $\Omega_{arag}$ conditions were seen during winter and in persistent tidal mixing zones, and corrosive $\Omega_{arag}$ values were also seen in areas that receive significant glacial melt in summer. The time-of-detection was computed and revealed that tidal mixing zones may be sentinel observing sites with relatively short time spans of observation needed to capture secular trends in seawater $pCO_2$ equivalent to the contemporary atmospheric $CO_2$ increase. Finally, anthropogenic $CO_2$ was estimated and showed notable time and space variability. We theoretically considered the change in hydrogen ion concentration ($[H^+]$), pH, and $\Omega_{arag}$ over the industrial era and to an atmospheric $pCO_2$ level consistent with a 1.5°C warmer climate and revealed greater changes in $[H^+]$ and pH in winter as opposed to larger $\Omega_{arag}$ change in summer. In addition, the contemporary acidification signal everywhere along the Inside Passage exceeded the global average, with Johnstone Strait and the Salish Sea standing out as potential bellwethers for biological OA impacts. In theory, roughly half the acidification signal experienced thus far over the industrial era may be expected over the coming 15 years with an atmospheric $CO_2$ trajectory that continues to be shaped by fossil-fuel development.

**Short summary** Information on the marine carbon dioxide system along the Inside Passage of the Pacific Northwest coast of North America has been limited. To address this gap, we instrumented an Alaskan ferry in order to characterize the marine carbon dioxide system. Data over a 2-year period were used to assess drivers of the observed variability, identify the timing of severe conditions, and assess the extent of contemporary ocean acidification as well as future levels consistent with a 1.5°C warmer climate.





# 1 Introduction

Atmospheric carbon dioxide ($CO_2$) has increased from 278 ppm in 1765 to 412 ppm in 2019 due to the emissions of
$CO_2$ from fossil fuel combustion and land use change, which combined have mobilized a total of 700 Gt of carbon that
otherwise would have remained locked in long-term geological reservoirs (Friedlingstein et al., 2020). So far over the industrial
era, an estimated 160 Gt of this carbon pool has transferred into the ocean, known as the oceanic anthropogenic $CO_2$ component
(Sabine et al., 2004), and has led to changes in the marine $CO_2$ system including reduced carbonate ion concentration ($[CO_3^{2-}]$) and pH, and increased hydrogen ion concentration ($[H^+]$) and $CO_2$ partial pressure ($pCO_2$). These marine $CO_2$ system changes
are collectively referred to as "ocean acidification" (Caldeira and Wickett, 2003;Doney et al., 2009;Feely et al., 2004a;Feely
et al., 2009), and two recent assessments estimate an average pH decline for the global surface ocean on the order of 0.1 units
(Jiang et al., 2019;Lauvset et al., 2020). In conjunction with this pH decline, reductions in $[CO_3^{2-}]$ have simultaneously
decreased the saturation state ($\Omega$) of carbonate biominerals; with aragonite being a more soluble carbonate biomineral that is
typically targeted in biological studies investigating the effects of ocean acidification (OA). $\Omega_{arag}$ is a ratio of the product of
$[CO_3^{2-}]$ and calcium concentrations over the solubility product for aragonite, and this ratio dictates the thermodynamic
favourability of aragonite precipitation. If $\Omega_{arag}$ is > 1, precipitation is favoured over dissolution. Globally, average surface
$\Omega_{arag}$ is estimated to have declined by 0.53 units (Lauvset et al., 2020). These assessments of global average pH and $\Omega_{arag}$
decline are based on calculations of anthropogenic $CO_2$ content, however, long-term change in both pH and $\Omega_{arag}$ resulting
from anthropogenic $CO_2$ input has been captured in prominent open ocean time series datasets (Bates et al., 2014;Doney et al.,
55   2020).

Along the continental margins, seawater conditions may not track the global average surface ocean pH decline,
particularly in the Northeast Pacific where seawater is less buffered than in some other ocean regions thereby making it more
sensitive to increasing anthropogenic $CO_2$ (Feely et al., 2018;Feely et al., 2008;Lauvset et al., 2020;Cai et al., 2020;Jiang et
al., 2015). One estimate of pH decline on this margin suggests twice the global average based on fossil foraminifera shells
preserved in marine sediments (Osborne et al., 2020). Even with the potential for larger pH decline along the Northeast Pacific
margin, putting this change into context can be challenging. Given that pH is negative $\log_{10}$ of $[H^+]$, the absolute change in
$[H^+]$ varies based on the initial pH for the same degree of pH decline (Fassbender et al., 2021;Fassbender et al., 2017). For
example, a 0.1-unit pH decrease with an initial pH of 7.6 will result in an absolute $[H^+]$ change of 6.3 nmol $kg^{-1}$, whereas the
same degree of pH decrease with an initial pH of 8.4 will drive a 1 nmol $kg^{-1}$ $[H^+]$ change. This clarification is important
because the absolute change in acidity can be different despite the same relative change in pH, and confusion may be enhanced
when considering that some continental margins likely have experienced different relative pH change compared to the global
surface ocean average (Osborne et al., 2020;Evans et al., 2019;Pacella et al., 2018;Salisbury and Jönsson, 2018) including in
some coastal ecosystems currently being evaluated for their OA mitigation potential (Ricart et al., 2021;Kroeker et al., 2021).

Evaluating the magnitude of change from OA and an ecosystem's mitigation potential are both critical areas of
research because negative impacts are already being felt by some vulnerable marine species. Along the Northeast Pacific





continental margin, larval shellfish mortality within hatcheries has been tied directly to low $\Omega_{arag}$ (Barton et al., 2012) and some adaptation measures to avoid such conditions have been developed (Barton et al., 2015). Other shell-forming marine species in this region are also exhibiting impacts from OA, including Dungeness crab (Bednarsek et al., 2020) and pteropods (Bednarsek et al., 2017;Bednarsek et al., 2021;Mekkes et al., 2021). The general consensus is that calcifying species may be the most directly impacted (Kroeker et al., 2013;Haigh et al., 2015;Marshall et al., 2017), although sensitivity to OA appears to be very species, life stage, and population specific (Doney et al., 2020) with the potential for compensatory mechanisms to help sustain populations (Peck et al., 2018;Bednarsek et al., 2021). However, there is the possibility of enhanced vulnerabilities by other co-occurring stressors like warming (Kroeker et al., 2013) and reduced oxygen content (Gobler and Baumann, 2016). Biological stressors, such as viral pathogens and harmful algal species, may also become more prevalent or virulent in association with changes in marine $CO_2$ chemistry and warming (Raven et al., 2020;Asplund et al., 2013). The sum of both the direct and indirect effects from OA and other co-stressors threatens marine food webs (Jin et al., 2020), harvested species (Ekstrom et al., 2015), and dependent coastal communities (Mathis et al., 2015); understanding this threat demands assessing how the marine $CO_2$ system has and will evolve through time.

Determining long-term trends in coastal settings is difficult because of inherent high variability resulting from a number of processes unique to the land-ocean interface. Physical forcing from upwelling-favourable winds or tide-induced vertical mixing can periodically result in surface water $pCO_2$ that is super-saturated with respect to the atmosphere, whereas high rates of primary production draw down surface water $pCO_2$ to well below atmospheric levels. Additionally, freshwater input from land can act to dilute total dissolved inorganic carbon ($TCO_2$) and total alkalinity and reduce $pCO_2$ (Meire et al., 2015), or, alternatively, increase $pCO_2$ through the respiration of riverine organic matter (Ward et al., 2017). These processes all occur on different time scales, are not uniformly important across coastal settings, and collectively act to make resolving the relatively small signal of anthropogenic $CO_2$ input difficult to disentangle from environmental variability. Resolving environmental variability, even to the point of capturing seasonal cycles, remains a challenge in many settings due to a lack of measurements (Hales et al., 2008). In the Northeast Pacific between British Columbia (BC) and southeast Alaska (AK), modelling efforts have aided in addressing this knowledge gap, and have indicated the relative significance of freshwater input (Siedlecki et al., 2017;Hauri et al., 2020) and its source character (Pilcher et al., 2016), as well as projected warming, deoxygenation, and acidification on multi-decadal time scales (Holdsworth et al., 2021). However, observations at appropriate time and space scales remain essential to evaluate model output and confirm our understanding of the governing processes that shape the variability, particularly in nearshore settings that are typically not well parameterized. The coast from BC to southeast AK has large under-sampled areas (Hales et al., 2008;Evans and Mathis, 2013) and coarse temporal information on marine $CO_2$ system variability based on direct measurements (Evans and Mathis, 2013;Evans et al., 2012;Tortell et al., 2012) except within the Salish Sea where seasonal and spatial patterns are somewhat more constrained (Evans et al., 2019;Cai et al., 2021;Ianson et al., 2016;Fassbender et al., 2018a;Feely et al., 2010;Lowe et al., 2019).

We reduced this information gap by outfitting a passenger ferry within the Alaska Marine Highway System (AMHS) fleet, the M/V *Columbia*, with instrumentation to monitor surface ocean conditions along a 1600-km track of the Pacific



Northwest coast known as the Inside Passage (Figure 1). The Inside Passage is a network of coastal waterways that spans the semi-enclosed Salish Sea, the central and northern BC coast, and the Alexander Archipelago of southeast AK. This nearshore region is a key interface between the Pacific coastal temperate rainforest (O'Neel et al., 2015;Bidlack et al., 2021) and a highly productive continental shelf ecosystem (Ware and Thomson, 2005;Jackson et al., 2015). The area experiences a wide array of physical and biogeochemical drivers including intense tidal currents within narrow passages that induce persistent vertical

mixing (Whitney et al., 2005;Dosser et al., 2021), strong autumn and winter storms (Stabeno et al., 2004), high runoff from rainfall and snow- and glacial-melt sources (Morrison et al., 2012;Beamer et al., 2016;Edwards et al., 2020;Neal et al., 2010), high terrestrial organic carbon input (Edwards et al., 2020;Oliver et al., 2017;St. Pierre et al., 2021), and remotely-forced influences such as El Niño events and marine heatwaves (Bond et al., 2015;Jackson et al., 2018).

We report here on surface underway measurements made from November 2017 to October 2019. Using this dataset,

we describe marine $CO_2$ system patterns along the Inside Passage, and quantify the relative importance of key drivers in shaping the observed $pCO_2$ variability. We further consider marine $CO_2$ system extremes and their timing along the ferry transit, which revealed differences based on how extremes were defined that likely has implications for the exposure histories of vulnerable species. Finally, we estimate the anthropogenic $CO_2$ content accrued over the industrial era and assess the impact this perturbation has had on $[H^+]$, pH, and $\Omega_{arag}$. We theoretically gauge the extent of acidification implied by the Paris

Agreement (UNFCC, 2015) to limit global warming to preferably 1.5°C relative to pre-industrial levels; what we refer to as the "1.5°C acidification level". The so-called "remaining 1.5°C carbon budget" translates to an atmospheric $CO_2$ level which would be reached with all potential mitigation pathways (Rogelj et al., 2018) and therefore can be viewed as the best case scenario for the maximum acidification owing to anthropogenic $CO_2$ input.

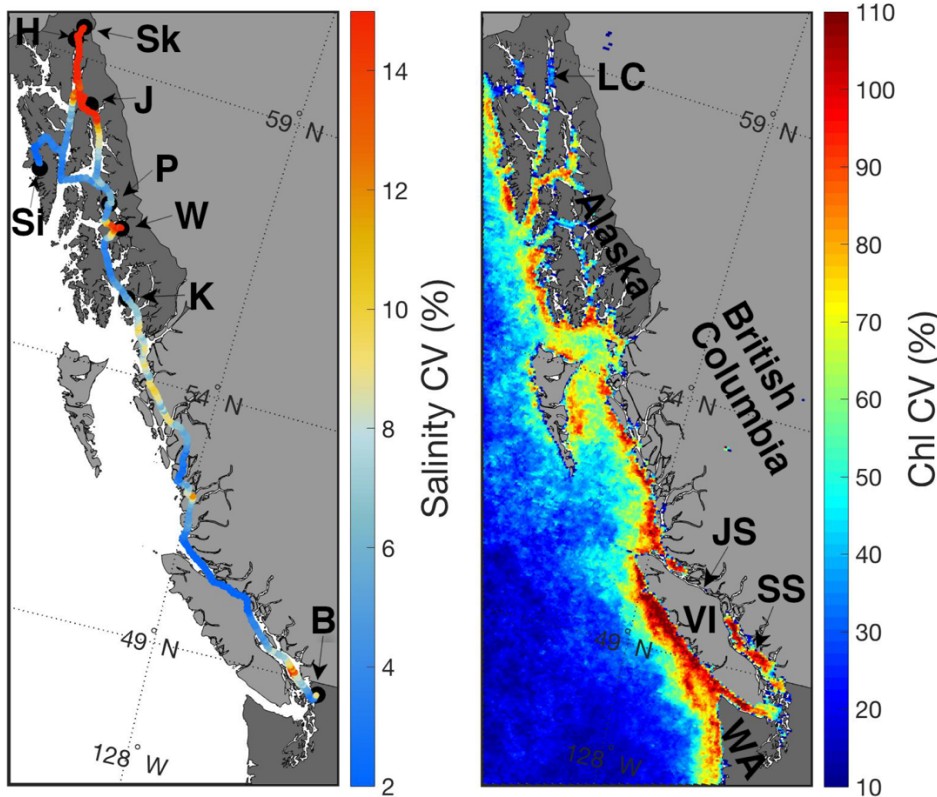

**Figure 1:** Surface salinity along the Inside Passage expressed as the coefficient of variation (CV; %; left) computed for 0.03° x 0.03° monthly grid cells from underway measurements made from the M/V *Columbia* between November 3, 2017 and October 2, 2019. Areas of highest salinity CV are due to large freshwater input, and black circles with labels mark the Alaska Marine Highway System terminals: Bellingham, WA (B), Ketchikan, AK (K), Wrangell, AK (W), Petersburg, AK (P), Juneau, AK (J), Haines, AK (H), Skagway, AK (Sk), and Sitka, AK (Si). Also shown is the surface chlorophyll CV (%; right) from Moderate Resolution Imaging Spectroradiometer (MODIS) Level 3 4-km mapped data from February to October 2018 and 2019. Areas of high chlorophyll CV reflect instances of biomass accumulation presumably owing to periods of high primary productivity. Areas labelled in the figure panel are: the states of Washington (WA) and Alaska, the province of British Columbia, Lynn Canal (LC), Johnstone Strait (JS), Vancouver Island (VI), and the Salish Sea (SS).

## 2 Methods

### 2.1 Underway Instrumentation

The AMHS M/V *Columbia* transited the 1600-km Inside Passage on a weekly basis. The vessel left Bellingham, WA (Figure 1) on Friday evening and arrived in Ketchikan, AK by Sunday morning. From there, the ship travelled north to



Wrangell, Petersburg, Juneau, Haines, and Skagway. After leaving Skagway, the *Columbia* travelled south to Sitka, then back to Ketchikan by Wednesday afternoon, and finally returned to Bellingham, WA by Friday morning. Maintenance on the instrumentation configured aboard the vessel was conducted during most visits to Ketchikan in order to prevent biofouling. Surface (~2 m) seawater $pCO_2$ data were obtained from measurements of $CO_2$ mixing ratio ($xCO_2$) made using a General Oceanics 8050 (GO8050) $pCO_2$ Measuring System following recommended protocols (Pierrot et al., 2009). Seawater was drawn into the M/V *Columbia* through an intake located in the bow thruster engine room and supplied to the GO8050 and ancillary sensors using a ½ HP self-priming centrifugal pump (AMT 429A-98 or similar) located ~2 m from the seawater intake. Temperature measurements were made at the seawater intake using an SBE 38 Digital Oceanographic Thermometer with an accuracy reported by Sea-Bird Electronics of 0.001°C. Seawater was then circulated from the bow thruster room up one deck to the car deck and then aft approximately 30 m along the starboard side to where the GO8050 and ancillary sensors were installed. The seawater circulation loop was split at this location between the GO8050 wet box and an ancillary sensor loop consisting of an SBE 45 MicroTSG Thermosalinograph and an Aanderaa 4330F oxygen optode. The accuracy of the temperature measurement from the SBE 45 was reported as 0.001°C when interfaced with the SBE 38, and the accuracy of the conductivity measurement was 0.0003 S/m. Salinity, computed from conductivity and temperature, is reported here on the Practical Salinity Scale (PSS-78). The accuracy of the Aanderaa 4330F oxygen optode reported by the manufacturer Xylem was < 1.5%. All ancillary sensors were serviced annually, and a multipoint calibration was conducted on the oxygen optode at the Aanderaa facility in Norway. Oxygen data from the Aanderaa 4330F was output in μmol l$^{-1}$, salinity-corrected using the approach described in Bittig et al. (2018), and then density-corrected to μmol kg$^{-1}$. Oxygen data are reported here as the difference from saturated values ($\Delta O_2$).

Seawater entered the GO8050 wet box at ~2.9 l min$^{-1}$ and ~10 psi, and then was circulated into a water-jacketed primary showerhead equilibrator with a liquid volume of ~0.5 l and to a smaller secondary equilibrator with a liquid volume of ~0.1 l. The primary equilibrator was maintained at ambient pressure on the car deck by a vent that was plumbed to the secondary equilibrator and then to the primary equilibrator. The pressure difference between inside the primary equilibrator and the car deck was monitored using a Setra pressure transducer (model 239) with a 0.15 hPa uncertainty. The secondary equilibrator serves to pre-equilibrate (make-up) air entering the primary equilibrator through the vent due to any loss through the headspace gas recirculation loop. A flow meter was present at the opening of the vent in order to monitor make-up air flow into the equilibrator. The headspace gas volume of the primary equilibrator was ~0.8 l, and seawater temperature was monitored within the primary equilibrator using a Fluke Thermometer (model 1523) and thermistor probe (model 5610) with an uncertainty of 0.01°C.

Atmospheric air was drawn from an intake on the foredeck to the GO8050 wet box. Both the equilibrator headspace gas and atmospheric air were dried using a condenser (Peltier thermoelectric cooling device) and Permapure Nafion drying tubes in order to minimize the correction for water vapor content associated with band-broadening within the infrared gas analyzer located in the GO8050 dry box. The analyzer housed in the GO8050 dry box was an LI-COR LI840A $CO_2$/$H_2O$ Gas Analyzer with a root-mean-square noise level for 1-Hz measurements of 1 ppm reported by the manufacturer. Dried





equilibrator headspace and atmospheric gases were supplied to the analyzer from the wet box at ~0.1 l min[-1]. In addition to the analysis of equilibrator headspace and atmospheric gases, four standard gases of known mixing ratio (150, 349, 449, and 850 ppm; Praxair) were also plumbed to provide gas flow to the GO8050. Praxair standard gases were evaluated by calibrating an

LI840A, and then using the calibrated analyzer to measure the $CO_2$ content of a World Meteorological Organization (WMO) traceable standard gas cylinder from the National Oceanic and Atmospheric Administration Earth System Research Laboratory (ESRL) Greenhouse Gas Global Reference Network. The Praxair gas standard calibrated LI840 was able to reproduce the certified ESRL standard to within 0.1%.

The GO8050 was controlled using National Instruments LabVIEW software run on a PC laptop computer. The

software controls data acquisition from the GO8050, an interface box connecting the SBE 38 and the SBE 45, the Aanderaa 4330F, the primary equilibrator temperature and pressure sensors, a Vaisala digital barometer (0.07 mbar accuracy) with a Model 61002 Gill Pressure Port and GPS antenna positioned adjacent to the atmospheric air intake, and the LI840A; while also controlling a Valco Instruments Co. Inc. (VICI) multi-port actuator that cycles between the gas streams plumbed to the dry box. The $CO_2$ measurement scheme controlled by the software involved the analysis of the four gas standards of known

$CO_2$ content, 12 analyses of atmospheric $CO_2$, and 240 seawater $CO_2$ measurements in a cycle that was repeated every 8.5 hours with a 2-min measurement frequency. The seawater and atmospheric $CO_2$ analyses were run in a sequence of 3 atmospheric measurements and 60 seawater measurements that was repeated 4 times between standardization. Analyses of each gas standard were interpolated over the time record of the dataset, and used to create calibration functions needed to correct the raw LI840A $xCO_2$ data. Calibrated seawater $xCO_2$ data in dry air were quality-controlled, and then converted to

$CO_2$ partial pressure ($pCO_2$) in wet air at the equilibrator temperature by using atmospheric pressure measured by the LI840A plus the differential pressure recorded in the equilibrator corrected for the removal of water vapor. Finally, seawater $pCO_2$ in wet air was adjusted to sea surface temperature using the offset between SBE 45 temperature recorded at the GO8050 and intake temperature from the SBE 38 located at the seawater intake with zero lag (0.3°C±0.17°C), as the lag between the temperature measurements at these two locations was determined to be less than the measurement frequency.

Total uncertainty in our $pCO_2$ measurements is the combined uncertainties from calibration, equilibrator temperature, equilibrator pressure, and the warming correction added in quadrature. At contemporary atmospheric $pCO_2$ levels near 400 µatm, these component uncertainties would equate to 0.4, 0.06, 0.17, 3.9 µatm. Typically, underway $pCO_2$ measurement uncertainties are reported as a function of uncertainties in the equilibrator temperature and pressure, and the water vapor pressure (Wanninkhof et al., 2013). Considering our dried gas stream that minimizes uncertainty from water vapor pressure,

the uncertainty from just the equilibrator temperature and pressure, with inclusion of the calibration uncertainty, would equal 0.44 µatm. However, taking into account uncertainty in the warming correction (while still not addressing deviations from a constant $pCO_2$ temperature sensitivity), increases the $pCO_2$ uncertainty to 3.92 µatm. While we prefer the more typical assessment that points to a lower $pCO_2$ uncertainty, we use a conservative ~1% $pCO_2$ uncertainty below to estimate the uncertainties in derived marine $CO_2$ system parameters.





205        In June 2019, a BioGeoChemical SUrface MOnitoring-system (BGC-SUMO) was configured with the GO8050 to provide underway pH measurements. The BGC-SUMO measures pH, temperature, and nitrate concentration, although the latter measurement was not successful on this vessel. The pH was measured using a Deep-Sea DuraFET, consisting of an Ion Sensitive Field Effect Transistor (ISFET) and a Chloride Ion-Selective-Electrode as the reference (Johnson et al., 2016). Thus, seawater is unmodified and no chemicals are added as it flows through the BGC-SUMO. The pH sensor was calibrated prior

to deployment on the M/V *Columbia*, and its performance was verified based on discrete samples taken alongside the sensor (n = 9) throughout the deployment (Takeshita et al., 2018). Based on this comparison, we assume an uncertainty in pH of 0.01. pH is reported here on the total hydrogen ion scale.

## 2.2 Discrete Sample Collection

        Discrete seawater samples were collected on two ferry trips in November 2017 and August 2018. Samples were drawn

from the seawater supply line immediately upstream of the GO8050 into rinsed 350 ml amber soda-lime glass bottles and analyzed for $TCO_2$ and $pCO_2$ within a month of collection following methods described elsewhere (Evans et al., 2019). Briefly, $TCO_2$ and $pCO_2$ were analyzed from the same sample bottle in this order at the Hakai Institute's Quadra Island Field Station using a Burke-o-Lator $pCO_2/TCO_2$ analyzer. The $TCO_2$ measurement was achieved by acidification and gas stripping followed by non-dispersive infrared detection using a LI-COR LI840A, and consumed ~60 ml of sample. $TCO_2$ measurements were

adjusted using correction factors developed through the analysis of certified reference materials (CRMs) from Andrew Dickson (Scripps Institute of Oceanography), with typical correction factors between 0.99 and 1.01. Uncertainty in the discrete $TCO_2$ measurement was determined to be 0.3% (Evans et al 2019). The $pCO_2$ measurement was achieved by headspace gas recirculation between the LI840A and the sample bottle in a closed loop until equilibration of the headspace gas with the seawater sample $pCO_2$ was obtained (roughly 6 minutes). Uncertainty in the discrete $pCO_2$ measurement was determined to

be 1%. The $TCO_2$ measurement was subsequently headspace gas corrected (Wanninkhof and Thoning, 1993), and then alkalinity (Alk) was computed using the $pCO_2$ and head space gas-corrected $TCO_2$ data with a MATLAB version of CO2SYS (Sharp et al., 2021) and the carbonic acid dissociation constants of Waters et al. (2014), the bisulfate dissociation constant of Dickson et al. (1990), the fluoride and hydrogen association constants from Perez and Fraga (1987), and the boron/chlorinity ratio of Uppström (1974). Alk consisted of $CO_3^{2-}$, bicarbonate, borate, hydroxide, and hydrogen ions and excluded

contributions from organic acids, phosphate, and silicate.

## 2.3 Calculations

### 2.3.1 Gap Filling, Marine $CO_2$ calculations, and Gridding

        Direct $pCO_2$ observations were compromised from August 25 to October 2, 2019. Subsequently, $pCO_2$ was determined indirectly using pH measurements and a regional Alk-salinity relationship (Evans et al., 2015). To fill these missing

data, pH measurements were interpolated to the measurement time of the GO8050. $pCO_2$ was then computed using the time-





matched pH data with the relationships described above and derived Alk. Missing measured $pCO_2$ observations during the period of compromised measurements in late 2019 were filled with the computed values.

Seawater pH, $[H^+]$, and $\Omega_{arag}$ were computed for the entire dataset using the salinity, intake temperature, the gap-filled $pCO_2$ record, and Alk derived from salinity (Evans et al., 2015) with the dissociation constants and relationships described
above using a MATLAB version of CO2SYS (Sharp et al., 2021). Uncertainty in pH, $[H^+]$, and $\Omega_{arag}$ derived from our $pCO_2$ record coupled with salinity-based Alk determinations was assessed using the error propagation routine from Orr et al. (2018) updated in the most recent MATLAB version of CO2SYS (Sharp et al., 2021). Combined standard uncertainties for pH, $[H^+]$, and $\Omega_{arag}$ were computed using the previously described 1% $pCO_2$ uncertainty, the reported 17.21 µmol kg$^{-1}$ uncertainty in the regional Alk-salinity relationship (Evans et al., 2015), and the default uncertainties for the dissociation constants within the
error propagation routine. pH, $[H^+]$, and $\Omega_{arag}$ uncertainties were computed across the range of observed Alk values and with $pCO_2$ computed across a range $TCO_2$:Alk ratios spanning 0.85 to 1 for each corresponding Alk value (Supplemental Figure S1). These calculations were done at a constant temperature and with salinity ranging from 10 to 32 corresponding with the range of Alk values. Importantly, marine $CO_2$ system data quality falls into two objectives as defined by the Global Ocean Acidification Observing Network (Newton et al., 2015;Tilbrook et al., 2019): (1) weather and (2) climate. The weather data
quality objective is thought sufficient for identifying spatial and temporal patterns excluding long-term trends, which is considered more appropriate for data reaching the stringent climate quality objective to assess. The mean pH, $[H^+]$, and $\Omega_{arag}$ uncertainties from our calculations are 0.01, 0.23, and 0.07 respectively. These values meet the Global Ocean Acidification Observing Network weather data quality objective. However, we note that uncertainties vary across the range of values considered. For instance, $\Omega_{arag}$ uncertainty is higher at higher $\Omega_{arag}$ values, whereas pH uncertainty is higher at lower salinity
and Alk (Supplemental Figure S1).

To evaluate basic statistics along the *Columbia* transit, including annual means, coefficients of variation (CV), and lower 5$^{th}$ percentiles, as well as assess seasonal drivers and the time-of-detection that are both described below, observations were gridded by isolating data within 0.03° by 0.03° grid cells. This grid size equalled roughly 6 km$^2$ across the latitudinal range of the *Columbia* transit. Analyses using gridded data was only conducted on grid cells containing more than 40
measurements.

### 2.3.2 Seasonal Drivers

The basis for assessing seasonal drivers stems from the following thermodynamic relationship from Takahashi et al. (1993):

$$\Delta pCO_2 = \left(\frac{\partial pCO_2}{\partial T}\right)\Delta T + \left(\frac{\partial pCO_2}{\partial TCO_2}\right)\Delta TCO_2 + \left(\frac{\partial pCO_2}{\partial TALK}\right)\Delta TALK + \left(\frac{\partial pCO_2}{\partial S}\right)\Delta S \qquad \text{(Equation 1)}$$

that defines the change in $pCO_2$ in seawater as a function of variation in temperature (T), $TCO_2$, total alkalinity (TALK, taken here to equal Alk, as defined above, when organic acid contributions are minimal), and salinity (S). Each partial differential





term represents a "buffer factor", with the two most commonly discussed in the literature being the Revelle factor (Sundquist

et al., 1979;Eggleston et al., 2010;Middelburg et al., 2020), $\left(\frac{\partial \ln pCO_2}{\partial \ln TCO_2}\right)$, and the temperature sensitivity (Takahashi

et al., 1993;Takahashi et al., 2002), $\left(\frac{\partial \ln pCO_2}{\partial T}\right)$. The global average Revelle factor is 10, meaning a 10% change in $pCO_2$

would result from a 1% change in $TCO_2$; whereas the temperature sensitivity of $pCO_2$ is 0.0423, meaning $pCO_2$ changes by

roughly 4% per degree temperature change. Processes that drive variation in these four terms are seasonal warming and

cooling, freshening and evaporation, physical transport and mixing, net community production (primary production minus

community respiration), sea-air $CO_2$ exchange, and calcification. The first two processes listed are thermodynamic drivers,

and the remaining processes are biophysical drivers. In most open ocean applications, seasonal variation in $pCO_2$ is dominated

by variation in temperature and $TCO_2$ (Takahashi et al., 2002). Removing the temperature component ($pCO_2$ T-component) of

the variability from the observations leaves behind variability driven by the remaining three terms in the above equation. $pCO_2$

variability with the $pCO_2$ T-component removed is expressed as:

$$pCO_2 \text{ at } T_{mean} = pCO_{2,obs} \times e^{0.0423(T_{mean} - T_{Obs})} \qquad \text{(Equation 2)}$$

where $T_{mean}$ is the annual mean temperature, and $pCO_{2,obs}$ and $T_{obs}$ are the $pCO_2$ and temperature (T) observations within a grid

cell. The $pCO_2$ T-component of seawater $pCO_2$ variability can be determined as the difference between the observations and

$pCO_2$ at $T_{mean}$. Here, the salinity component was computed using the relationship from Sarmiento and Gruber (2006) derived

from Equation 1:

$$pCO_2 \text{ S} - \text{component} = \Delta S \times \left(\frac{\overline{pCO_2}}{\overline{S}}\right) \times \left(\gamma_S + \gamma_{TCO2} + \gamma_{TALK}\right) \qquad \text{(Equation 3)}$$

that importantly captures the impact of changing salinity on not just $pCO_2$ sensitivity (the last term in Equation 1) but also

$TCO_2$ and Alk. This is done using annual mean $pCO_2$ and salinity (S) in each grid cell, defined as $\overline{pCO_2}$ and $\overline{S}$, with the salinity

($\gamma_S$), $TCO_2$ ($\gamma_{TCO2}$), and TALK ($\gamma_{TALK}$) buffer factors (Takahashi et al., 1993). We use global average values for these three

buffer factors in this computation. Note that $\gamma_{TCO2}$ and $\gamma_{TALK}$ oppose each other, and differences from the global averages

would be largely compensated for by the competing influence of increasing (or decreasing) $TCO_2$ and increasing (or

decreasing) Alk. Finally, the thermodynamic components, $pCO_2$ T-component and $pCO_2$ S-component, can be combined and

then differenced from the observations to isolate variability that results from the remaining biophysical drivers. The seasonal

amplitude of each component of $pCO_2$ variability was assessed, and either the difference or ratio of the amplitude of

thermodynamic (T, S, or TS) to biophysical drivers ($B_T$, $B_S$, $B_{TS}$; where subscript denotes the removed terms) defines which

is more important for determining $pCO_2$ variability on an annual basis (Takahashi et al., 2002;Fassbender et al., 2018b).

### 2.3.3 Severity and Time-of-Detection



We determined the severity of derived pH and $\Omega_{arag}$ in each grid cell based on the lower 5th percentile as in Chan et al. (2017), and the timing of severe conditions as the mode of all months of observations less than or equal to the lower 5th

percentile of each grid cell. We also assessed the time-of-detection (ToD) within each grid cell of the *Columbia* transit in order to guide future observational efforts targeting the identification of long-term change. ToD is similar to the time-of-emergence used in climate studies (Henson et al., 2017) with the exception that it includes measurement uncertainty (Carter et al., 2019b). Both of these terms represent the time required for a secular trend, in our case increasing seawater $pCO_2$ from anthropogenic $CO_2$ uptake, to emerge from the "noise" in an environmental dataset. Monthly mean $pCO_2$ is computed from the observations

occurring within each grid cell, and then the observations are differenced from the monthly mean in order to compute de-seasonalized anomalies (*i.e.,* removing the large amplitude seasonal cycle from the "noise"). The standard deviation of the de-seasonalized anomalies was combined in quadrature with the $pCO_2$ measurement uncertainty to represent the remaining environmental "noise" and compute ToD as:

$$\text{ToD} = \frac{2 \times \text{noise}}{pCO_2 \text{ growth rate}} \qquad \text{(Equation 4)}$$

where the $pCO_2$ growth rate used here was 2.5 µatm yr$^{-1}$. Importantly, the growth of seawater $pCO_2$ can vary across coastal settings and may or may not be entirely driven by anthropogenic $CO_2$ input (Laruelle et al., 2018). For example, changes in nutrient input from runoff can alter the $pCO_2$ growth rate from an expected anthropogenic $CO_2$ driven signal (Turk et al., 2019). Therefore, we present ToD only to discuss how these data might be used to target observing efforts and not as absolute values.


### 2.3.4 Anthropogenic $CO_2$

Anthropogenic $CO_2$ content was estimated using the $\Delta TCO_2$ approach (Takeshita et al., 2015; Pacella et al., 2018; Evans et al., 2019), which is a simplification of the $\Delta C^*$ method (Sabine et al., 2002; Gruber et al., 1996), and assumes a constant $TCO_2$ disequilibria with the atmosphere defined as:

$$\Delta TCO_{2,\text{diseq}} = TCO_{2,\text{obs}} - TCO_2\left(\text{atm } pCO_{2,\text{current year} - \text{age}}, \text{Alk}_{\text{der}}, T_{\text{obs}}, S_{\text{obs}}\right) \qquad \text{(Equation 5)}$$

where $TCO_{2,\text{obs}}$ is the observed $TCO_2$ and $TCO_2\left(\text{atm } pCO_{2,\text{current year} - \text{age}}, \text{Alk}_{\text{der}}, T_{\text{obs}}, S_{\text{obs}}\right)$ is the $TCO_2$ content that would result from equilibration with the atmospheric $pCO_2$ at the time of last contact with the atmosphere (current year minus the age of the water mass), at the derived Alk, and at the observed temperature and salinity. The time of last contact with the atmosphere represents the age of a water mass in years, and is zero for most surface measurements except in areas where deep

water is mixed to the surface. Water mass age was estimated by dividing the measured apparent oxygen utilization (AOU, or the inverse of $\Delta O_2$) by the oxygen utilization rate (OUR). A value of 4.1 µmol kg$^{-1}$ yr$^{-1}$ for OUR was taken from the literature for Pacific subarctic upper water (Feely et al., 2004b) and used in this calculation. Using the $\Delta TCO_{2,\text{diseq}}$ term and assuming patterns in derived Alk and observed temperature and salinity are largely invariant, the $TCO_2$ for a given year can be estimated by:





$$TCO_{2,year} = TCO_2\left(atm\ pCO_{2,year-age}, Alk_{der}, T_{obs}, S_{obs}\right) + \Delta TCO_{2,diseq} \qquad \text{(Equation 6)}$$

where $TCO_{2,year}$ is the $TCO_2$ content for a specific year and $TCO_2\left(atm\ pCO_{2,year-age}, Alk_{der}, T_{obs}, S_{obs}\right)$ is the $TCO_2$ content that would be realized if that surface water mass were in equilibrium with the atmospheric $pCO_2$ that occurred during a given year, corrected for the age of the water mass, and at the contemporary derived Alk and observed temperature and salinity. The anthropogenic $CO_2$ content is then determined as the difference between the $TCO_2$ for a given year and the $TCO_2$ content for

the year 1765. Historical atmospheric $CO_2$ mole fractions based on observations and projected atmospheric $CO_2$ for the shared socio-economic pathways (SSP) were obtained from Meinshausen et al. (2020) using their data portal (http://greenhousegases.science.unimelb.edu.au) and converted to $pCO_2$ assuming standard atmospheric pressure. pH, $[H^+]$, and $\Omega_{arag}$ were computed for each year from 1765 onward using the $TCO_2$ estimated for a given year with the modern derived Alk and observed temperature and salinity. It is important to acknowledge that uncertainty in estimating anthropogenic $CO_2$

content using this approach is at least 5 µmol kg$^{-1}$ based on similarities with the $\Delta C^*$ method (Sabine et al., 2002). Uncertainty stems from a number of sources, including the key assumptions of constant $TCO_2$ disequilibria and unchanged variation in the natural carbon cycle, temperature, and salinity. Inadequacies in these assumptions can lead to biases in anthropogenic $CO_2$ (Matsumoto and Gruber, 2005), which in term influences estimations of past and future pH, $[H^+]$, and $\Omega_{arag}$.

## 3 Results and Discussion

### 3.1 Time and Space Variability

Over 244,000 seawater temperature, salinity, O$_2$, and pCO$_2$ measurements were made on the M/V *Columbia* during 135 north- and south-bound transits of the Inside Passage over a 2-year period. These data revealed substantial time and space variability in surface seawater conditions along this 1600-km stretch of coastline. The spatial and temporal mosaic captured by these measurements (Figures 2 and 3) portrays two key features of the Inside Passage: (1) the dominant mode of temporal

variability is the seasonal cycle, and (2) there is regional variability in the seasonal cycle amplitude that is modulated by the relative influences of tidal mixing, net community production, and the magnitude and character of freshwater input. Below we describe how these points first apply to the observations of sea surface temperature (SST) and salinity, and then to oxygen and the marine CO$_2$ system.

Between November and March, cold seawater spans the entire Inside Passage, with coldest water in the Alexander

Archipelago generally near 4°C but 0.5°C was observed near Juneau. Seasonal warming in most regions began in April and occurred earlier in the Salish Sea, which was consistent with satellite observations that have identified earlier seasonal warming in this region relative to coastal areas to the north (Jackson et al., 2015). Surface salinity was fresher throughout the year in the Salish Sea, although variability in salinity was larger in the Alexander Archipelago (Figures 1 and 2) where seasonal freshwater delivery to the coastal ocean contributes 41% of the freshwater input to the Gulf of Alaska (Edwards et al., 2020). The

combined estimates of discharge from each major watershed along the Inside Passage from Edwards et al. (2020) and Morrison





et al. (2012) indicate that over 570 km$^3$ yr$^{-1}$ of freshwater enters the Northeast Pacific from southeast AK, an amount that exceeds the Mississippi River discharge (Dai and Trenberth, 2002). Along the BC portion of the Inside Passage, discharge is near 390 km$^3$ yr$^{-1}$ with almost a quarter of this amount originating from the Fraser River. Despite lower runoff from BC, its influence on salinity manifests earlier than the peak freshwater input in southeast AK (Figure 2) due to the high contribution

of snow-melt to the late spring and early summer discharge (Morrison et al., 2012). In southeast AK, seasonal reduction in salinity began in May and reached the summer minima in August. Low salinity conditions were uniform over a large area that encompassed Lynn Canal and the inside waterways around Juneau (Figures 1 and 2). The late summer minima in salinity reflects the significant contribution of glacial melt in the more northern portion of the Inside Passage (Neal et al., 2010;Edwards et al., 2020). Seasonal variation in temperature and salinity was reduced in some confined waterways, such as Johnstone Strait

(Figure 1), owing to the influence of intense tidal mixing in these areas that dampens the seasonal cycle (Dosser et al., 2021;Whitney et al., 2005).





**Figure 2:** Time (x) – longitude (y) – latitude (z) plots of underway sea surface temperature (SST; °C; top) and salinity (bottom) collected between November 3, 2017 and October 2, 2019.


Across most of the Inside Passage, $\Delta O_2$ and $pCO_2$ showed an inverse relationship (Figure 3). Where $\Delta O_2$ values were positive, $pCO_2$ was undersaturated with respect to the atmosphere, and this combination reflects primary productivity exceeding rates of organic matter respiration; *i.e.*, positive net community production (NCP). However, we acknowledge that $\Delta O_2$ represents the combination of biotic and abiotic drivers, as does $pCO_2$. Abiotic changes in $\Delta O_2$ can result from changes

in temperature and salinity as well as bubble injection and wave breaking (Juranek et al., 2019), although we contend that the



latter two drivers may be of lesser importance in the protected Inside Passage waterways. Seasonal warming can increase both $\Delta O_2$ and $pCO_2$. However, we observed an increase in $\Delta O_2$ with a corresponding decrease in $pCO_2$, which was a strong indication that $O_2$ supersaturation and $pCO_2$ drawdown resulted from positive NCP (Tortell et al., 2012;Juranek et al., 2019). In areas outside of the influence of tidal mixing, the signals of $O_2$ supersaturation and $pCO_2$ drawdown initiated in response to

the spring phytoplankton bloom, and generally were sustained through summer until autumn storm season commenced (Evans et al., 2019;Fassbender et al., 2018a). An exception was Lynn Canal in the northern Alexander Archipelago (Figure 1) where the relationship between $O_2$ and $pCO_2$ diverged in summer (Figure 3) when the seasonal change in salinity was maximal (Figure 2). Cold glacial melt water results in undersaturated surface $pCO_2$ (Cai et al., 2021;Pilcher et al., 2016;Evans et al., 2014) while also increasing oxygen solubility and subsequently decreasing $\Delta O_2$ (Figure 3). The diverging character between

$O_2$ and $pCO_2$ in Lynn Canal dissipated during autumn when salinity increased in response to storm-induced vertical mixing. Autumn marked the transition back to supersaturated $pCO_2$ with respect to the atmosphere along the entire transit. Inter-annual variability was also apparent in this dataset during the spring and summer months, as 2019 had stronger $O_2$ supersaturation and $pCO_2$ drawdown during the spring bloom in the Salish Sea and on the central BC coast, and throughout much of the summer in the Alexander Archipelago (Figure 3).



**Figure 3:** $\Delta O_2$ (µmol kg$^{-1}$; top) and pCO$_2$ (µatm; bottom) as a function of longitude (x), latitude (y), and time (z) from November 3, 2017 to October 2, 2019.

Seasonal $O_2$ supersaturation and pCO$_2$ drawdown does not occur uniformly along the Inside Passage, but in distinct regions separated by areas of tidal mixing that support sustained low-$O_2$ and high-pCO$_2$ conditions (Figure 3) due to the near-continuous ventilation of sub-surface waters (Whitney et al., 2005;Dosser et al., 2021;Evans et al., 2012;Tortell et al., 2012). The most obvious region of intense tidal mixing along this coastline was in Johnstone Strait between Vancouver Island and





mainland BC (Figure 1), but other areas were also evident including in the narrow waterway north of Sitka (Figure 1) known as Sergius Narrows. As mentioned above, the seasonal amplitude in the tidal mixing zones is reduced because the water column

may be completely mixed and seasonal variation in these areas may more reflect that of sub-surface water entering the mixing zone laterally (Dosser et al., 2021). Since seasonality in these areas is potentially more influenced by sub-surface source waters, the seasonal cycle can be out of phase with adjacent areas outside of the tidal mixing zones. This was most obvious in Johnstone Strait, where high-$pCO_2$ conditions outside of this area were generally seen during winter, whereas within this region, highest $pCO_2$ was in autumn. The highest observed seawater $pCO_2$ was near 1200 $\mu$atm in Johnstone Strait during September. Winter

$pCO_2$ values outside of the tidal mixing zones broadly ranged between 450 and 800 $\mu$atm, being higher in regions with less direct connection to the open continental shelf, such as in the semi-enclosed Salish Sea, in areas of the central BC coast, and in the Alexander Archipelago (Figure 3). These areas receive high amounts of riverine organic matter (St. Pierre et al., 2021;Oliver et al., 2017;Johannessen et al., 2003) that may be confined to the nearshore zone by winter-time downwelling circulation (Thomson, 1981;Weingartner et al., 2009), and there subsequently remineralized by the microbial community (St.

Pierre et al., 2020) leading to elevated nearshore surface $pCO_2$ that is not seen in the offshore data along this coast (Evans and Mathis, 2013). In tidal mixing zones like Johnstone Strait, highest $pCO_2$ in early autumn decreased through winter to a minimum by late spring, albeit with values that were still supersaturated with respect to atmospheric $pCO_2$. This difference in timing reflects the seasonality of ventilated sub-surface waters (Dosser et al., 2021), as, without a short residence time (Pawlowicz et al., 2007), these waters would experience a build-up of respiratory $CO_2$ through the growing season as organic

matter rains out of the surface layer and is respired at depth by the microbial community. We suggest this sub-surface respiration signal is ventilated in the tidal mixing zones, and is responsible for the early autumn peak in surface $pCO_2$.

Seawater pH and $\Omega_{arag}$ variability was evaluated by employing an Alk-salinity relationship that was developed using data spanning a large portion of this coastline (Evans et al., 2015). Validation of this relationship was done using Alk determined from seawater samples collected during ferry ride-along cruises and processed as described above. These cruises

occurred in November 2017 and August 2018, and spanned the dynamic range of observed salinity conditions (Figure 2). During November, comparison between discrete Alk and salinity-derived Alk was within two times the root-mean-square-error of the salinity-based relationship (Supplemental Figure S2). During August, larger divergence between discrete and salinity-derived Alk occurred in low salinity water at the northernmost portion of the Inside Passage. Specifically in the area of Lynn Canal, Alk determined from the salinity-based relationship overpredicted bottle-determined Alk by at most 200 $\mu$mol

kg$^{-1}$. Salinity-based Alk determination was further evaluated in 2019 by comparing estimated pH, computed from directly-measured $pCO_2$ and salinity-based Alk, to directly-measured pH (Supplemental Figure S3). pH was measured from June to October 2019 over the period of lowest observed salinities in Alexander Archipelago, and revealed a similar pattern to the discrete Alk comparison. Divergence between estimated and directly-measured pH was greatest in seawater with salinity < 22 and north of 57°N in the region around Juneau and up Lynn Canal (Supplemental Figure S3). In the analysis that follows, we

continue to use salinity-based Alk with our gap-filled $pCO_2$ record to determine components of the marine $CO_2$ system along the Inside Passage, but acknowledge that the northernmost region during summer is likely more corrosive for aragonite than



our analysis suggests because of our Alk over-predictions in low salinity water. While the magnitude of how corrosive the $\Omega_{arag}$ values are in this region may be less well-constrained, the drivers and timing of adverse conditions should not deviate from what we describe below. The variance in freshwater Alk end-members in this region demands further study in order to

more accurately assess the magnitude of corrosive summer $\Omega_{arag}$ conditions in these glacial-melt influenced waters.



**Figure 4:** $pH_T$ (total scale; top) and $\Omega_{arag}$ (bottom) as a function of longitude (x), latitude (y), and time (z) from November 3, 2017 to October 2, 2019.

Patterns in seawater pH and $\Omega_{arag}$ were largely the inverse of that for $pCO_2$ (Figure 4); areas exhibiting $pCO_2$ undersaturation with respect to the atmosphere typically co-occurred with high pH and $\Omega_{arag}$ conditions, whereas regions with high $pCO_2$ have low pH and $\Omega_{arag}$. Areas with both high pH and $\Omega_{arag}$ have experienced recent positive NCP that would also support $O_2$ supersaturation and $pCO_2$ drawdown (Figures 3 and 4). The evidence of inter-annual variability discussed above for spring and summer $O_2$ and $pCO_2$ was apparent for pH and $\Omega_{arag}$, with 2019 exhibiting more frequent occurrences of $\Omega_{arag}$

> 3 compared to 2018. pH and $\Omega_{arag}$ was lowest in most areas during winter and year-round within tidal mixing zones. Winter $\Omega_{arag}$ values were < 1 in all regions that lacked direct connection to the open continental shelf; specifically, within the Salish Sea, Johnstone Strait, inside passages on the central and northern BC coast, and in Alexander Archipelago. Corrosive conditions for aragonite persisted throughout the year in Johnstone Strait, and in an area known as Wrangell Pass between Wrangell and Petersburg (Figure 1). In the northern Alexander Archipelago, a short period of $\Omega_{arag}$ conditions > 1 occurred

between March and June, resulting from the spring phytoplankton bloom as evidenced by co-occurring $O_2$ supersaturation and $pCO_2$ drawdown (Figure 3). Once the summer melt season commenced, the Inside Passage-wide minima in $\Omega_{arag}$ was observed in this region despite the over-prediction in Alk in low salinity water mentioned above. Lynn Canal exhibited the most corrosive conditions for aragonite along the 1600-km *Columbia* transit due to the large contribution of glacial melt in this region (Figure 2). Such corrosive conditions in glacial melt-influenced settings have been reported previously in AK

(Reisdorph and Mathis, 2013;Evans et al., 2014) as well as in Svalbard (Ericson et al., 2019;Cantoni et al., 2020). Co-occurring corrosive conditions for aragonite (Figure 4) and undersaturated $pCO_2$ with respect to the atmosphere (Figure 3) are unique to cold, glacial-melt influenced coastal regions, which could enable positive feedback whereby $CO_2$ influx from the atmosphere either enhances or prolongs corrosive summer $\Omega_{arag}$ conditions (Evans et al., 2014;Ericson et al., 2019;Cantoni et al., 2020).

### 3.2 Seasonal Drivers

$pCO_2$ variability is determined by thermodynamic and biophysical forcings; the latter being the sum of the physical and biogeochemical influences of vertical mixing, horizontal transport, NCP, sea-air $CO_2$ exchange, and calcification. Seasonal variation in $pCO_2$ reflects the interaction of these terms, which often are competing. For instance, seasonal warming and freshwater input have opposing influences on $CO_2$ solubility such that together they can dampen $pCO_2$ variability (Cai et al., 2021). As described above, the influence of seasonal warming and freshwater input can be assessed by perturbing the mean

$pCO_2$ by the seasonal change in temperature and salinity (Takahashi et al., 2002;Sarmiento and Gruber, 2006). The ratio of the seasonal amplitude of $pCO_2$ at $T_{mean}$ (B; Supplemental Figure S4) and the $pCO_2$ T-component (T; Supplemental Figure S5) provides information on whether biophysical processes or seasonal warming are more important for shaping $pCO_2$ variability within a region. Takahashi et al. (2002) describe this as T/B (or as a difference, T-B, see their Figure 9), where if T/B (or T-





B) is greater than 1 (or positive), seasonal temperature change is the dominant process determining $pCO_2$ variability. The global analysis by Takahashi et al. (2002) suggests that in the area nearest BC and southeast AK, temperature and biophysical processes play equal roles in determining $pCO_2$. A more recent analysis by Fassbender et al. (2018b) produced similar results for the northeast Pacific with balanced roles of temperature and biophysical processes evident nearest the coast. However, both of these analyses were conducted with large global grids that did not resolve the coastal margin and did not differentiate the role of freshwater given the open-ocean focus.

In the nearshore zone spanning BC and southeast AK, freshwater input exceeds 900 km$^3$ yr$^{-1}$ (Morrison et al., 2012;Neal et al., 2010;Edwards et al., 2020;Beamer et al., 2016), and the associated changes in salinity are substantial within specific regions (Figures 1 and 2). In such settings, it is essential to account for salinity variation when assessing nearshore $pCO_2$ variability. As pointed out by Sarmiento and Gruber (2006), variations in $pCO_2$ that result from changes in salinity cannot be evaluated based solely on the salinity sensitivity (Takahashi et al., 1993) because this only accounts for changes in solubility, and not the corresponding change in $TCO_2$ and Alk from a decrease in salinity. Instead, the contribution of changes in salinity to the $pCO_2$ variability can be evaluated by incorporating $TCO_2$ and Alk buffer factors into the calculation (see Sarmiento and Gruber 2006 Equation 10.4.2 or Eveleth et al., 2016). Changes in salinity would result from both freshwater input (decrease) and vertical mixing (increase), and are expressed here as the $pCO_2$ S-component (S; Supplemental Figure S6). As mentioned above, the $pCO_2$ S-component and $pCO_2$ T-component can be in opposition; when seasonal warming is maximal, freshwater content can also be maximal, and their corresponding influences on $pCO_2$ can be counterbalanced (Supplemental Figure S7). However, there are times and locations when these factors are not balanced. Lynn Canal (Figure 1) during the summer months is an important example of an area and time period when salinity variability exceeds the influence of seasonal warming (Figure 5). Subtracting both the $pCO_2$ S-component and the $pCO_2$ T-component (TS) from the observed $pCO_2$ leaves remaining variability associated with NCP, calcification, and gas exchange (Supplemental Figure S8). Given that calcification is only episodically important in this region and gas exchange is generally slow (order months), this remaining $pCO_2$ variability should largely reflect the influence of NCP, or $CO_2$ removal and addition by organic matter production and degradation, respectively.





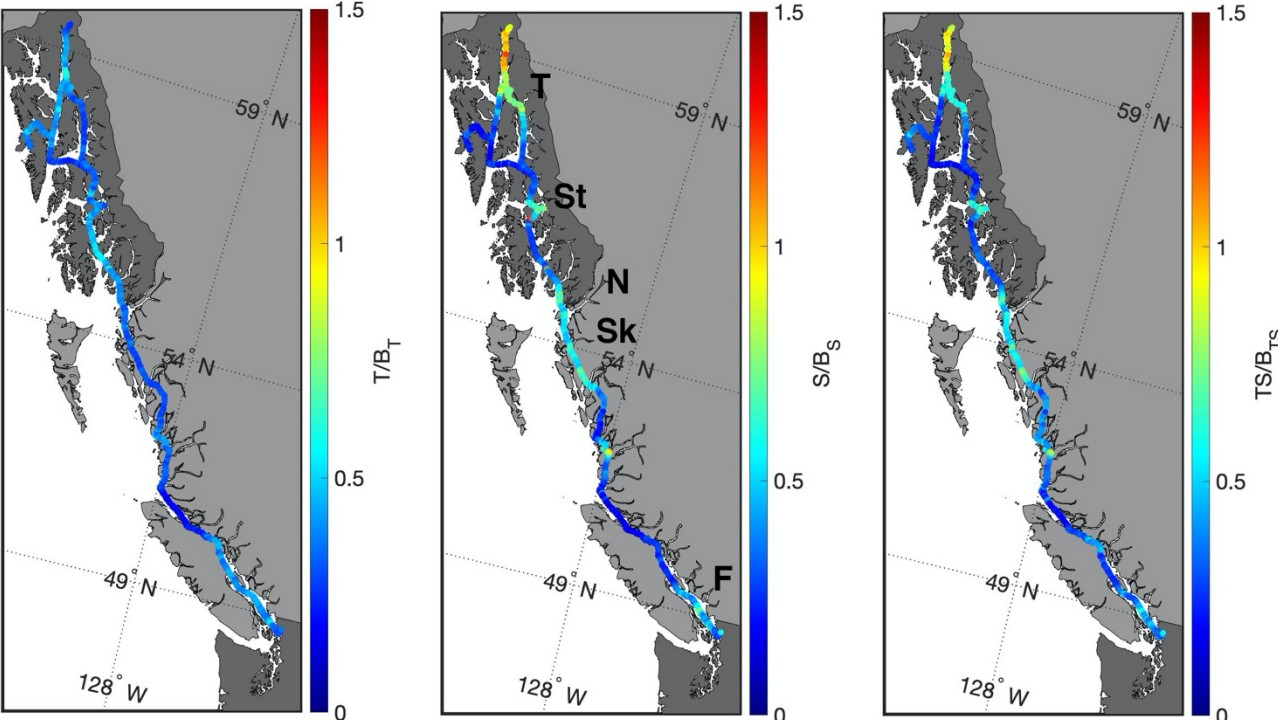

**Figure 5:** Ratios of the seasonal amplitude of thermodynamic and biophysical drivers of pCO$_2$ variability. The left panel is the ratio of the amplitude of the temperature component (T) to the amplitude of the remaining biophysical components (B$_T$). Middle panel is the ratio of the amplitude of the salinity component (S) to the amplitude of the remaining biophysical components (B$_S$). Major rivers are highlighted in this panel (Taku River (T), Stikine River (St), Nass River (N), Skeena River (Sk), and the Fraser River (F)). The right panel is the ratio of the amplitude of the combined temperature and salinity components (TS) to the amplitude of the remaining biophysical components (B$_{TS}$).

As illustrated in Figure 5, the biophysical component dominates over the temperature component in shaping pCO$_2$ variability on an annual basis everywhere along the Inside Passage. Excluding Lynn Canal, the salinity component is also less important than the biophysical component, even in areas adjacent to major river outflows. At the outflows of major rivers, such as the Fraser and Stikine (Figure 5), the salinity component is an important contributor but still roughly 30% less than the amplitude of the biophysical component. In Lynn Canal, salinity variance exceeded all other locations along the Inside Passage (Figures 1 and 2), which resulted in a dominant contribution to the pCO$_2$ variability (Figure 5). Temperature counterbalanced some of the salinity component in Lynn Canal, such that this was the only area where there was near equivalence between the combined thermodynamic components and the biophysical drivers in determining pCO$_2$. These computations show the spatial complexity in the balance between thermodynamic and biophysical drivers in the nearshore zone, and that the influence of salinity must be considered with temperature in such settings with significant freshwater input. The importance of salinity in





shaping marine $CO_2$ system variability in this region has been discussed previously in modelling studies by Siedlecki et al. (2017) and Hauri et al. (2020), as well as by Pilcher et al. (2016) who evaluated the role of variability in freshwater Alk end-members in enhancing nearshore atmospheric $CO_2$ uptake. However, the contributions of thermodynamic versus biophysical drivers to the observed variability has not been evaluated to the extent shown here, which indicated the dominance of
biophysical drivers over most of the Inside Passage.

The northernmost reach of the Inside Passage is heavily influenced by changes in salinity resulting from the volume of glacial melt water entering this area (Neal et al., 2010;Edwards et al., 2020). Reisdorph and Mathis (2013) first described the influence of melt water on marine $CO_2$ chemistry in this region, and subsequent observational and modelling work has considered the apparent de-coupling that can occur between $pCO_2$ and $\Omega_{arag}$ in locales of significant cold glacial melt discharge
(Evans et al., 2014;Ericson et al., 2019;Hauri et al., 2020;Cantoni et al., 2020). Given that atmospheric $CO_2$ uptake is promoted in glacially-influenced regions, these areas may be important amplifiers of OA (Cantoni et al., 2020;Ericson et al., 2019;Evans et al., 2014). Increasing glacial discharge, changes in glacial melt-water Alk as glaciers further recede and the flow path over land to the ocean increases, increasing glacial river temperatures, and increasing organic matter decomposition in glacial rivers are all factors that would modulate the extremely corrosive conditions within these impacted nearshore environments, as well
as the decoupling between $pCO_2$ and $\Omega_{arag}$. Given the potential for intensifying positive feedback with further increasing atmospheric $pCO_2$ and enhanced sea-air $CO_2$ exchange, thereby amplifying the already extreme $\Omega_{arag}$ conditions, additional research effort should target these areas in order to understand which feedbacks are most important from seasonal to inter-annual time scales.

### 3.3 Characterizing Regional Extremes

Identifying regional extremes in the marine $CO_2$ system is important for characterizing environmental variability, identifying where unfavourable conditions for vulnerable marine species occur more often or more intensely, and pin-pointing areas that may experience faster rates of change from anthropogenic $CO_2$ input (Feely et al., 2018;Hare et al., 2020). Marine $CO_2$ system extremes were characterized here based on pH and $\Omega_{arag}$ variability and severity (Chan et al., 2017). Importantly, these two descriptors for extremes may not manifest the same way in a region or with the same timing; a region may have
severely low pH but also low variability, and experience severely low pH at a different time of year than in adjacent areas. pH and $\Omega_{arag}$ extremes can also exhibit differences within a region.



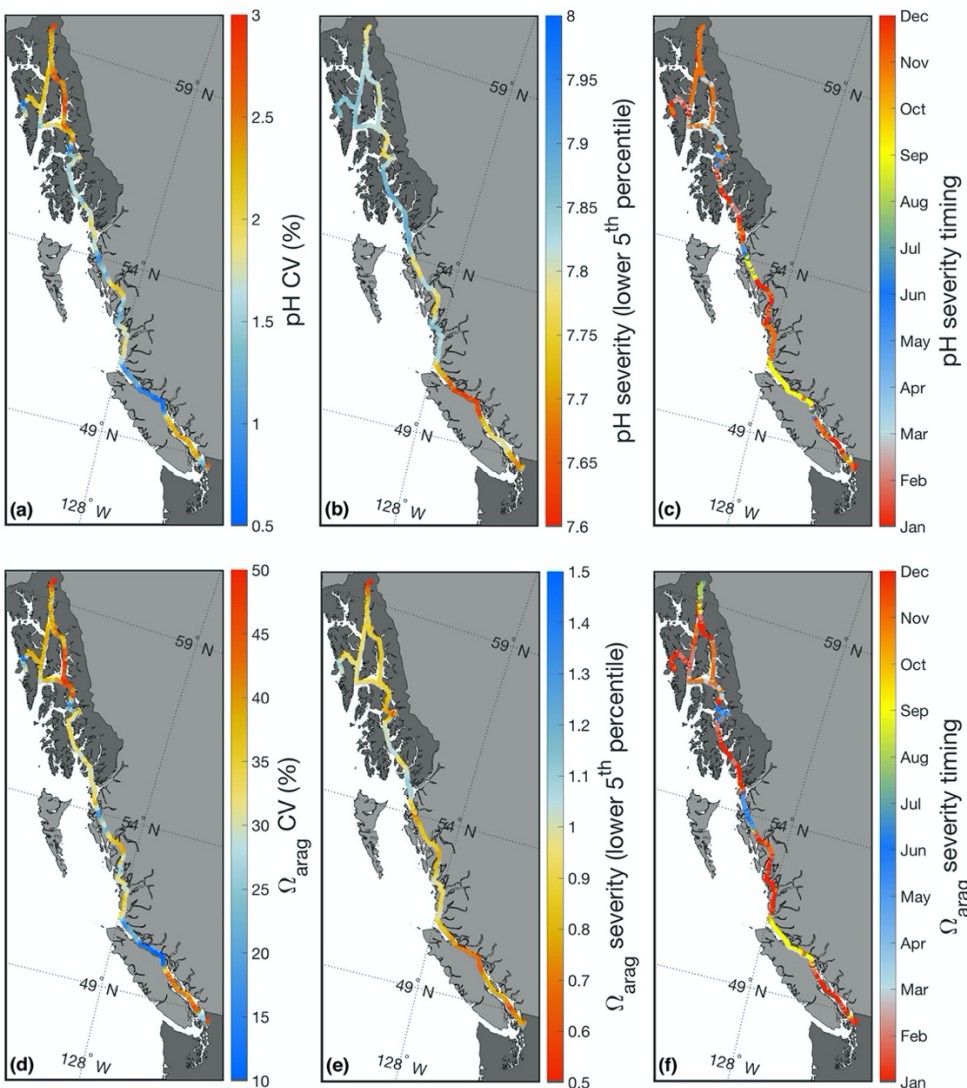

**Figure 6:** Panels a, b, and c show the pH coefficient of variation (CV), severity, and the timing of severe pH conditions, respectively. Lower panels d, e, and f show the same parameters for $\Omega_{arag}$.


  Extremes, as defined by the variability, were regionally similar for pH and $\Omega_{arag}$; both the Salish Sea and select areas within the Alexander Archipelago exhibited large variability relative to other areas along the Inside Passage (Figure 6). Contrasting these highly variable areas, Johnstone Strait, Sergius Narrows, and Wrangell Pass all had low variability owing to the influence of persistent tidal mixing. Using severity to portray extremes provided a nearly inverse picture, with Johnstone

550  Strait exhibiting both severe pH and $\Omega_{arag}$, while Lynn Canal had severe $\Omega_{arag}$ values but less severe pH. Notably, $\Omega_{arag}$ severity was only above 1 in surface water most exposed to the open North Pacific between AK and BC, an area known as Dixon





Entrance, and in Sitka; areas exposed to the open continental shelf generally had less severe and variable pH and $\Omega_{arag}$ compared to confined waters along the Inside Passage.

There were also differences in the timing of severe pH and $\Omega_{arag}$ conditions across and within regions (Figure 6). The

majority of Inside Passage waters experienced severe pH and $\Omega_{arag}$ between November and February when seawater $pCO_2$ was highest, however, severe conditions occurred in some areas earlier in the year. In Johnstone Strait, most severe pH and $\Omega_{arag}$ occurred in September, whereas in the areas near the Skeena and Stikine outflows (Figure 5), most severe $\Omega_{arag}$ occurred in June and likely in association with the snow-melt freshet. In Lynn Canal, most severe $\Omega_{arag}$ values were in August due to the peak input of glacial melt, and in November for pH owing to storm-induced vertical mixing. This temporal variation in most

severe conditions may have biological implications because of the potential for synchronized timing with the occurrence of more sensitive life stages of vulnerable species. In addition, the different characterization of extremes based on variability and severity potentially impacts adaptation trajectories as, for example, vulnerable organisms in Johnstone Strait would experience a sustained corrosive and moderately stable low pH environment; whereas in the Alexander Archipelago, vulnerable organisms would be subjected to large swings in marine $CO_2$ system parameters over the year. While some research considers long-term

exposure to variable marine $CO_2$ conditions as a factor enhancing physiological tolerance to OA (Kapsenberg and Cyronak, 2019), other research suggests that organisms living in persistently low pH environments might be more locally adapted (Chan et al., 2017). Here we provide information on the locations of both of these types of settings such that future work can move to examine how species fare along this gradient within the Inside Passage.




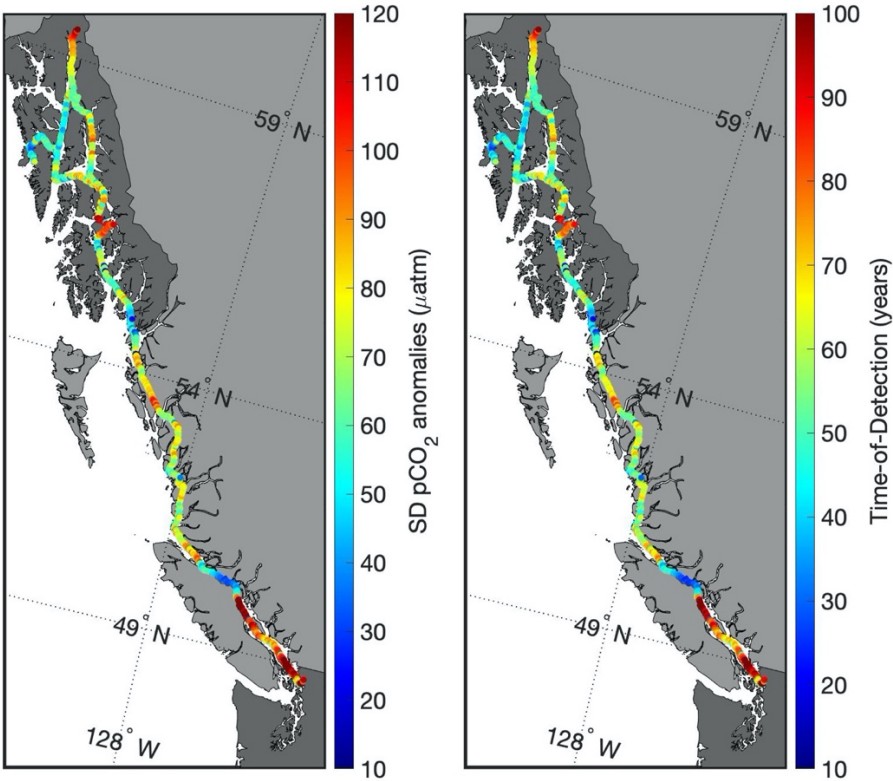

**Figure 7:** Standard deviation of pCO₂ anomalies (μatm; left) and the time-of-detection (years; right) to resolve the signal of
increasing seawater pCO₂ that tracks the contemporary atmospheric CO₂ trajectory.

Variability-based characterization of extremes is also useful for optimizing observing efforts by ensuring assets are
deployed in locations that either minimize the amount of time anticipated to observe an OA-driven change in the marine CO₂
system or capture a key process that may be driving large amplitude signals in a region (Turk et al., 2019), which may itself
be impacted by climate change. Areas with low natural variability require fewer years to resolve a secular trend as opposed to
regions of high variability (Sutton et al., 2019). Figure 7 shows the standard deviation of de-seasonalized pCO₂ observations
(anomalies), representing the environmental "noise" along the Inside Passage, with the resulting ToD (computed following
Equation 4). The Salish Sea, the area near the Stikine River, and the northern portion of Lynn Canal all have very long ToD,
and observing efforts in these regions would be better suited for targeting the processes discussed previously that shape the
variability. As these processes may themselves be subject to climate change (Bidlack et al., 2021), tracking their evolving
influence on marine CO₂ system parameters will provide valuable information on the dynamics organisms are subjected to in
these highly variable environments. On the other hand, Johnstone Strait, the area near Ketchikan, and Sergius narrows near
Sitka all have much shorter ToD (Figure 7). These areas would be ideal for placing observing assets aimed at resolving long-
term secular trends. It is also important to understand that the ToD estimates computed here are "forced" values (Turk et al.,



2019), in that they are based on seawater $pCO_2$ increasing at the same pace as the present atmospheric $CO_2$ increase (Sutton et al., 2019). In some cases, there may be an observed trend in a time series that differs from the "forced" trend (Laruelle et al., 2018), and this can reflect either the role of other processes independent of anthropogenic $CO_2$ increase that modulates the trend in seawater $pCO_2$ (Turk et al., 2019) or faster increases in $pCO_2$ resulting from anthropogenic $CO_2$ addition in weakly-buffered settings (Feely et al., 2018). Areas, such as Johnstone Strait, with low variability that results in short ToD, and with

severe pH and $\Omega_{arag}$ stemming from higher $TCO_2$:Alk ratios and weaker buffering, likely will exhibit faster rates of change than estimated by "forced" ToD values. Identifying and establishing these areas as sentinel sites for tracking OA would optimize coastal observing efforts aimed at resolving long-term secular trends.

## 3.4 Estimating Past and Future Conditions

We consider below how the marine $CO_2$ system along the Inside Passage has evolved over the industrial era as well

as what additional change we might anticipate if our society is able to reduce greenhouse gas emissions to reach the preferable Paris Agreement target of 1.5°C warming (UNFCC, 2015); referred to here as the 1.5°C acidification level. It is important to note that our evaluation is theoretical and only considers the role of anthropogenic $CO_2$, and not the influence of other forcings like increasing temperature or changing freshwater input. Following the approach outlined above, anthropogenic $CO_2$ for Inside Passage surface waters was determined and showed notable spatial and temporal variability (Figure 8). This variability

was strongly influenced by freshwater input and water mass age, of which the latter ranged from 0 to 35 years (Supplemental Figure S9). Maximal age estimates were confined to the areas of persistent tidal mixing, and were similar to estimates from other studies for the age of upwelled water present on the Northeast Pacific continental shelf (Feely et al., 2008;Murray et al., 2015). This agreement is encouraging considering the potential for inaccurately representing OUR in the calculation due to a likely higher oxygen utilization in the confined near shore regions (Johannessen et al., 2014;Pawlowicz et al., 2007). However,

it is worth considering how inaccurately estimating water mass age translates to uncertainty in anthropogenic $CO_2$ and characterization of preindustrial $[H^+]$, pH, and $\Omega_{arag}$. Inaccurate water mass age estimates are more influential in older water masses at the surface and over time periods when atmospheric $pCO_2$ is changing faster (i.e., in computing contemporary $\Delta TCO_{2,diseq}$). Keeping this in mind, we consider how a 50% uncertainty in the age of surface water in Johnstone Strait impacts our estimation of preindustrial conditions. An overestimate of the age by 50% results in a lower $\Delta TCO_{2,diseq}$ by 11.2 µmol kg$^{-1}$

$^{1}$, and a higher anthropogenic $CO_2$ by 11.6 µmol kg$^{-1}$. This overestimate changes computation of the contemporary $[H^+]$, pH, and $\Omega_{arag}$ acidification signals (i.e., the difference between contemporary and preindustrial values) by 0.96 nmol kg$^{-1}$, 0.03, and -0.08, respectively. An underestimate of the age by 50% would increase $\Delta TCO_{2,diseq}$ by 9.6 µmol kg$^{-1}$ and decrease anthropogenic $CO_2$ by 10 µmol kg$^{-1}$. The contemporary $[H^+]$, pH, and $\Omega_{arag}$ acidification signals would adjust by -1.4 nmol kg$^{-1}$, -0.04, and 0.07, respectively. Despite the presence of variability in the age estimate for Johnstone Strait (Supplemental Figure

S9), we suggest it is unlikely that the age is underestimated. Rather, this variability is more likely a function of seasonal





variation in sub-surface water mass oxygen utilization (Johannessen et al., 2014) and the application of a constant OUR; although, given that the presence of older, upwelled water is seasonal along this coastline (Feely et al., 2016), some variability in the age of surface water masses within persistent tidal mixing zones is expected. We therefore use the shifts in $[H^+]$, pH, and $\Omega_{arag}$ values resulting from an overestimate of the water mass age by 50% as uncertainty bounds when considering the

contemporary and 1.5°C acidification levels below.

        Interestingly, areas identified in the previous section as pH and $\Omega_{arag}$ extrema based on severity, due to either persistent tidal mixing or glacial melt input, were not locales with the highest anthropogenic $CO_2$ content. Instead, highest values were in regions that experienced the greatest $O_2$ supersaturation and $pCO_2$ draw down during summer (Figures 3 and 8). Highest estimated values were near 66 µmol kg$^{-1}$ and similar to those reported for coastal surface water in other recent North Pacific

studies (Carter et al., 2019a;Feely et al., 2016), however the lower values in fresher seawater have, to our knowledge, not been reported. The Salish Sea is considered to have more moderate anthropogenic $CO_2$ levels (Feely et al., 2010;Hare et al., 2020;Evans et al., 2019), but the freshest areas observed in the Alexander Archipelago exemplify unique settings with very low anthropogenic $CO_2$ content and where conditions have likely been corrosive for aragonite over most of the industrial era. We estimated the first year when $\Omega_{arag}$ was < 1 along the Inside Passage (Figure 8) by accounting for the anthropogenic $CO_2$

content accrued each year over the industrial era. Weakly-buffered areas, either due to being very fresh or because of ventilation of sub-surface water with high $TCO_2$ relative to Alk, were shown to have likely always been corrosive at least on a seasonal basis. These naturally corrosive hot spots have been amplified by anthropogenic $CO_2$ addition such that, for example, Johnstone Strait now experiences under-saturation on an annual basis (Figure 4). This also implies that the sub-surface water ventilated in Johnstone Strait was previously corrosive to aragonite on a seasonal basis. The shift to corrosive winter conditions occurred

over more recent decades outside of mixing zones and areas impacted by glacial melt. Winter surface water in the Salish Sea likely transitioned to $\Omega_{arag}$ < 1 beginning around 1950, consistent with the emergence of corrosive winter values found in a previous study (Evans et al., 2019). Similar winter transition timing was evident for Inside Passage waters on the central BC coast (Figure 8). Within the Alexander Archipelago, the winter transition to $\Omega_{arag}$ < 1 likely appeared more recently over the last few decades.




**Figure 8:** Contemporary anthropogenic $CO_2$ content ($\mu$mol kg$^{-1}$) and the estimated first year when $\Omega_{arag}$ was < 1.

650          Matthews et al. (2021) determined the allowable future $CO_2$ emissions that limits global warming to the preferred 1.5°C level stated in the Paris Agreement (UNFCC, 2015), the so-called "remaining carbon budget", to be 440 GtCO$_2$ from 2020 onwards. Using the relationships described in Friedlingstein et al. (2020), full emission of the remaining carbon budget can be equated to a rise in atmospheric $CO_2$. With 3.664 GtCO$_2$ equalling 1 GtC, and every 2.124 GtC of emissions increasing





atmospheric $CO_2$ by 1 ppm, the remaining carbon budget would drive an increase in atmospheric $CO_2$ of 56 ppm. At 1

atmosphere of pressure, combining the 2019 atmospheric $CO_2$ mole fraction with the remaining carbon budget-forced atmospheric $CO_2$ increase results in an atmospheric $pCO_2$ of 468 μatm. The time at which this atmospheric $pCO_2$ would be realized is trajectory-dependent, with both the sustainable development pathway (SSP1) and the fossil-fuel development pathway (SSP5) reaching this atmospheric $pCO_2$ level at very different times (Meinshausen et al., 2020). SSP5 reaches, and surpasses, this value quickly, by roughly 2035. SSP1 takes longer and reaches this level by 2063. Here we use this atmospheric

$CO_2$ target to consider the theoretical 1.5°C acidification level, and report the year 2035 as the fastest trajectory following the central estimate for the year when 1.5°C warming would be reached if the current rate of warming continues (IPCC, 2018). However, what is important is that this is an anticipated extent of OA reachable with either trajectory. If our society holds to SSP1, this is theoretically the most acidification we should expect without considering amplifying processes. However, if we follow a SSP5-type scenario, acidification will surpass what we estimate below.







**Figure 9:** The contemporary [H$^+$] values minus the estimated values for 1765 (top) and the estimated values for 2035 minus the 1765 values (bottom).

The estimated extent of contemporary acidification along the Inside Passage is, on average, a 38% increase in [H$^+$] over the 254 years since the start of the industrial era, although important spatial and temporal variability in acidification was evident (Figure 9). This average extent of acidification equates to a 0.14 unit drop in pH. The Inside Passage average pH





change exceeds the global average (Lauvset et al., 2020;Jiang et al., 2019), however, similar to [H⁺], pH change varied spatially and temporally with values ranging from 0.06 to 0.20 (Supplemental Figure S10). Spatially, [H⁺] and pH change has been greatest in the more weakly-buffered and moderately anthropogenic CO₂-concentrated waters of the Salish Sea and Johnstone

Strait. The largest [H⁺] change was evident in Johnstone Strait (Figure 9), with a maximum near 7 nmol kg⁻¹, due to the inherently lower background pH level (Fassbender et al., 2021). For comparison, open ocean surface water has experienced an average [H⁺] change of 1.6 nmol kg⁻¹ (Fassbender et al., 2021). The pH change in these two settings appeared only marginally different, being 0.17 versus 0.1, but with an over 4-fold increase in [H⁺] in Johnstone Strait. This highlights that making comparisons to global average surface ocean pH changes in coastal settings can be potentially misleading. Temporally,

and despite a higher anthropogenic CO₂ signal during summer (Figure 8), the change in [H⁺] and pH appeared larger in winter. Note that the seasonal variation in the acidification signal exceeded the 0.96 nmol kg⁻¹ and 0.03 [H⁺] and pH uncertainties, respectively, discussed previously based on inaccuracies in the water mass age estimate. This seasonality in acidification manifests because of seasonal differences in TCO₂:Alk ratios (Supplemental Figure S11) that alters the CO₂ system response to anthropogenic CO₂ increase. During winter, the TCO₂:Alk ratio is closer to unity such that seawater is more weakly-buffered

and the percent change in pH per unit change in TCO₂ has a greater magnitude. Larger percent changes in pCO₂ and $\Omega_{arag}$ are also expected during this season, and this pattern follows the modelled (Fassbender et al., 2018b;Kwiatkowski and Orr, 2018) and observed (Landschützer et al., 2018) changes in the seasonality of surface marine CO₂ parameters at the global scale.

At an atmospheric pCO₂ of 468 µatm, an additional 17% increase in [H⁺], on average, would theoretically be expected for the 1.5°C acidification level. This implies that roughly half of the acidification experienced thus far over the industrial era

will likely occur over the coming 15 years if society maintains the current emissions trajectory. However, a change in emissions trajectory that follows a sustainable development pathway would enable this acidification signal to occur over a longer period of time. It is anticipated that acidification will be further amplified during winter along the Inside Passage and particularly within the more weakly-buffered semi-enclosed waterways (Figure 8). Johnstone Strait and the Salish Sea will likely continue to experience the largest changes in [H⁺] (Figure 9). Along the Inside Passage, these areas may serve as bellwethers for

biological OA impacts in a similar manner as how high-latitude settings are viewed (Fabry et al., 2009). Efforts to examine biological impacts *in situ* should target these regions where we estimate the largest contemporary and 1.5°C acidification levels. In addition, studies challenging organisms to adverse [H⁺] and pH levels within experimental settings may benefit from our estimates of the 1.5°C acidification level, as this could serve as a near-term treatment for diagnosing OA impacts.




**Figure 10:** The contemporary $\Omega_{arag}$ values minus the estimated values for 1765 (top) and the estimated values for 2035 minus the 1765 values (bottom).


     Unlike for pH, change in $\Omega_{arag}$ over the industrial era has a seasonal maximum in summer (Figure 10). This characteristic can also be explained by considering the seasonality in seawater $TCO_2$ and Alk as well as relative versus absolute changes in $\Omega_{arag}$. Despite the $TCO_2$:Alk ratio being closer to unity during winter, and the percent change in $\Omega_{arag}$ per unit change





in TCO$_2$ being larger during that season (Supplemental Figure S11), summer $\Omega_{arag}$ values are much higher than winter values

(Figure 4). A 14% change in an $\Omega_{arag}$ value near 1 is a smaller absolute change than a 9% change in an $\Omega_{arag}$ value of 3. Considering the 1.5°C acidification level, the change in summer $\Omega_{arag}$ values at times may exceed 0.8 units, consistent with an overall reduction in seasonality as anthropogenic CO$_2$ content continues to increase (Kwiatkowski and Orr, 2018). To our knowledge, differences in the season during which maximum changes in [H$^+$] and pH versus $\Omega_{arag}$ occur has not been widely acknowledged in the literature, and points to the need for careful consideration of the specific marine CO$_2$ system parameter

an organism may be most sensitive to (Waldbusser et al., 2014). Seasonally-specific changes in the most impactful marine CO$_2$ system parameter for a sensitive species may or may not align with periods of maximal vulnerability (Hales et al., 2016). Considering how the marine CO$_2$ system is being modified by increasing anthropogenic CO$_2$ on a seasonal as well as long-term basis, and which specific variable is most impactful for an organism, are both essential elements for understanding the implications of OA.

**4 Conclusions**

Through partnership with the Alaska Marine Highway System, we have reduced the information gap on marine CO$_2$ system variability along the Inside Passage of the Pacific Northwest coast of North America. This study has documented the time and space variability in surface water along this 1600-km passageway, shown that the dominant mode of temporal variability is the seasonal cycle, and that the amplitude of this signal is modulated by the relative influences of tidal mixing,

net community production, and the magnitude and character of freshwater input. We have highlighted that variability in freshwater alkalinity end-members in the northern Inside Passage demands further study given the potential for positive feedback with atmospheric CO$_2$ uptake and for modifications in freshwater outflows that can alter the coastal OA signals. The analysis of seasonal drivers in our study indicated that the biophysical component has a dominate role in shaping variability along most of the Inside Passage, but that the combined influences of temperature and salinity balance the biophysical

component in the northernmost region. We considered the characterization of pH and $\Omega_{arag}$ extremes, and recognized that there are regional differences in the manifestation of extremes based on variability versus severity that likely have biological implications. Vulnerable organisms experiencing a sustained corrosive and moderately stable low pH environment may have a differing adaptation trajectory than organisms subjected to large swings in marine CO$_2$ system parameters over the year. Our diagnosis of these locations should be useful for future studies examining adaptation and climatization. We also used our

variability assessment to determine the time-of-detection and point out that this information can help optimize coastal observing efforts aimed at resolving long-term secular trends. Finally, we estimated the anthropogenic CO$_2$ content in surface water and considered change over the industrial era as well as to an atmospheric CO$_2$ level that would corresponds with the exhaustion of the remaining 1.5°C carbon budget. Through these calculations, we estimated the year when $\Omega_{arag}$ was first suppressed below a value of 1. It is likely that some areas, including the tidally mixed Johnstone Strait and in the northernmost



portion of Lynn Canal, have experienced seasonal $\Omega_{arag}$ values < 1 over the entire industrial era. Other areas have transitioned to winter $\Omega_{arag}$ values < 1 more recently. Evaluated contemporary acidification levels revealed seasonal differences in the changes in $[H^+]$, pH and $\Omega_{arag}$, with $[H^+]$ and pH changes being larger during winter when conditions are more weakly-buffered while $\Omega_{arag}$ change is larger during summer. Looking to the future, we considered the 1.5°C acidification level and estimated significant marine $CO_2$ system changes over the coming 15 years if society continues on a fossil-fuel development emissions

trajectory.

**Code availability**

MATLAB routines developed as part of this study are available upon request to the corresponding author.

**Data availability**

Three datasets were generated through this study: (1) the record of directly-measured surface $pCO_2$, (2) the gap-

filled $pCO_2$ record including measurements of pH from the BGC-SUMO, and (3) the measurements of $TCO_2$ and $pCO_2$ on discrete samples collected during the ferry ride-along cruises. The directly-measured surface $pCO_2$ record can be found within the Surface Ocean $CO_2$ Atlas data holdings (https://www.socat.info/) as well as within the Ocean Carbon and Acidification Data Portal at the National Centers for Environmental Information (https://www.ncei.noaa.gov/data/oceans/ncei/ocads/metadata/0209049.html). The gap-filled $pCO_2$ record including BGC-

SUMO pH data can be found with the discrete sample dataset in the Hakai Institute's data portal (https://doi.org/10.21966/m0es-7520).

**Author Contribution**

WE and AB procured State of Alaska Department of Transportation approval for the equipment installation aboard the M/V Columbia and the funding for this project. WE, GL, and CH oversaw the installation and operation of the GO8050

$pCO_2$ Monitoring System and ancillary sensors. YT, WE, and CH oversaw the installation and operation of the BGC-SUMO. WE participated in ride-along cruises and collected discrete samples for validation. WE conducted the analysis and wrote the manuscript. All authors contributing to revising and editing the manuscript for submission.





**Competing Interests**

The authors declare that they have no conflict of interest.

**Disclaimer**

Nothing in this report is endorsed by or reflects the views of the State of Alaska Department of Transportation and Public Facilities.

**Acknowledgement**

We gratefully acknowledge funding support from the Alaska Ocean Observing System, the Alaska Coastal Rainforest Center at the University of Alaska Southeast, and the Tula Foundation. YT, and work at the Monterey Bay Aquarium Research Institute, was supported by the David and Lucile Packard Foundation and NSF OCE-1736864. This project was made possible through partnership with the State of Alaska Department of Transportation, and we thank the crew of the M/V *Columbia* who helped to maintain the integrity of the dataset. We also thank Katie Pocock and Carrie Weekes for processing the discrete samples used to assess the regional Alk-salinity relationship. This is PMEL contribution number 5298 and CICOES contribution number 2021-1157.

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
