# Peer review of "Marine CO2 system variability along the Northeast Pacific Inside Passage determined from an Alaskan ferry"

_Biogeosciences, 2021_

## Author Response (AR1)

Reviewer 1:

Review of Evans et al. "Marine $CO_2$ system variability along the Inside Passage of the Pacific Northwest coast of North America determined from an Alaskan ferry."

Evans et al. have prepared a massive manuscript detailing a suite of data collected opportunistically over two years in the Pacific Northwest. The core observations are temperature, salinity, and $pCO_2$; however, the authors leverage a salinity-based estimate of total alkalinity (TA) to expand the analysis to other parameters of interest such as DIC, pH and â ¦. This approach is clever and allows for some novel analysis of anthropogenic influences on the marine inorganic carbon system in this region, but has drawbacks which are highlighted. Speaking from my own experience, I appreciate that continuous data records from ships of opportunity can be challenging to assemble into a coherent scientific analysis. I applaud the authors' work here in that regard, and think this work achieves that end. This manuscript is well-written, but some details in the Methods might be moved to the Supplementary Material. There is a huge amount of variability in study region conditions, so some specific sub-regions of interest are highlighted. This regional variability is presented in a series of nice figures, but some statistics around months/seasons might help to reinforce the presentation of the data, perhaps in a couple tables.

MAJOR COMMENTS

-I am somewhat concerned with the length. For some journals this manuscript would be too long by perhaps 25%. The Methods section takes up more than 25% of the total manuscript text. These methods are important, but some of the description could be presented in the Supplementary Material. Perhaps Biogeosciences is a good fit, as being online-only the length is not a publication concern, but I do think readability would be helped with some length reduction and more concision.

That you for this comment. We have worked to condense the introduction, added a "Study Region" section, condensed some of the Methods including moving the description of calculating seasonal drivers to the Supplemental Material as suggested by the reviewer, and condensed portions of the discussion section.

-Many of the Figures (2-4, 8 and 9) follow the same presentation style, showing the cruise track repeated over time and colored by data values. This is a style I haven't seen often before and overall I think it is really effective. However, I can't figure out the time component of these plots. For example, in Figure 2 each panel has dates listed along the x-axis (I'm going to ignore the longitude axis at the left side of these plots for now). The x-axis tick mark corresponds to Nov17, which agrees with the Introduction text as to when surveys began. However, as the ship travels 'north' (vertically along the y-axis), it also travels west, but seemingly back in time as well. Thus, the survey that departed Nov17 heading north appears to arrive at the northernmost point in Skagway a couple months earlier. This becomes confusing when seasonality is discussed: while the total north-south transit took one week, the northernmost data appear to precede the southernmost data by a couple months, even though the north-south total transit took one week. One solution might be to add a secondary x-axis on the top of each panel, corresponding to the date when the ship arrived at the northernmost point (basically the lower x-axis shifted to the left). These plots also seem to be done in a Matlab 3-D format (with short longitude z-axes appearing at the bottom),

but I'm not sure the inclusion of the longitude adds much (although longitude is probably necessary to generate the plot). Can they be presented in a more 2-D format, or can the figure captions be expanded to provide more detail to the reader?

We thank the reviewer for this comment. We maintain the 3-D structure of the figures because we believe this is important for representing the cruise track in southeast AK where it deviates from a mainly N-S orientation, and because this format provides a coastline and terminal positions that can be used to reference back to Figure 1. Therefore, we have edited the legends of these figures to better orient the reader to the 3 axes.

MINOR COMMENTS

-The title itself is long. Could it be something like "Pacific Northwest marine $CO_2$ system variability along the Inside Passage coast"?

Thank you, we have shortened the title to: Marine CO2 system variability along the Northeast Pacific Inside Passage determined from an Alaskan ferry

-L41-42: is 1765-present considered the industrial era?

Thank you, we have clarified the definition of industrial era in the Lines 41-43.

-L46: 0.1 decline over what period?

Thank you, we have added "over the industrial era".

-L52-55: not sure what this is getting at

We have clarified this point that records of direct measurements of pH or omega are not long enough to constrain trends over the industrial era, but that the multidecadal time series that do exist show declining pH and omega.

-L69: change in what?

We have replaced "change" with "marine CO2 system changes"

-L71: was this mortality linked to upwelling of potentially anthropogenic $CO_2$?

Thank you, yes, the low omega conditions were linked to ocean acidification and the provided reference (Barton et al., 2012) reflects this.

-L96-98: variability of what? This sentence may be so general as to be unnecessary.

We have clarified "marine CO2 system" variability in the sentence.

-L117-118: awkward phrasing here

Thank you, we have revised this statement entirely in the new "Study Region" section.

-Figure 1: the arrows are hard to see in some cases. Can they be outlined in white, or made wider? Also, can full location names (instead of acronyms) be used in the map itself?

Thank you, we have increased the width of the arrows to make them more visible. We have also added Sergius Narrows, Wrangell Narrows, and Dixon Entrance to the map following a comment made by the reviewer below. However, we maintain the use of acronyms because otherwise the map would be crowded with text.

-L134: I know I've mentioned the length of the manuscript, but a section here laying out the basic geography of the study area would be useful, especially since the discussion leans heavily on some specific geographic/oceanographic characteristics like areas of stronger tidal mixing and freshwater input.

Following this comment, as well as a similar comment from Reviewer 2, we have added a "Study Region" section for these details.

-L157: what is the water jacket for? Temperature control?

The water-jacket around the primary equilibrator is used to minimize warming. A clarifying phrase has been added to this statement.

-L170: "a LI-COR"

Corrected.

-L175: "calibrating a LI840A using the Praxair gases, then using…."

Corrected.

-L184-186: So was $CO_2$ measured an about a 2-minute interval? Were other data (SST, salinity) also measured or recorded at 2 minutes?

Thank you, this point was clarified by saying "The software captured measurements from all ancillary sensors as well as analyses of the four gas standards of known $CO_2$ content, 12 measurements of atmospheric $CO_2$, and 240 seawater $CO_2$ measurements in a cycle that was repeated every 8.5 hours with a 2-min measurement frequency.".

-L229-230: I'm not sure if the statement about the Alk composition is true, or even needed. CO2SYS certainly uses Alk contributions from sulfate and fluoride in the determination of total Alk. A sentence about the possible effects of the presence of organic alkalinity, P or Si could illustrate the potential uncertainties from leaving these potential contributors out. I believe modified versions of CO2SYS are available that can model organic alkalinity inputs.

Thank you for this comment. We have adjusted the statement to read "Alk computed in this way excluded contributions from organic acids, phosphate, and silicate.".

-L233: what compromised the $pCO_2$ data?

Thank you, we have clarified the first few sentences of Section 3.3.1 to explain the data gaps, following the comment below, as well as describe that the direct pCO2 measurements were compromised due to an issue with the LI-COR during late 2019.

-L234: "$pCO_2$ was estimated indirectly…"

Corrected.

L248-255: Not sure the weather vs. climate distinction matters much, as the authors clearly present the actual uncertainty estimates

Thank you for this comment. We maintain the description of weather vs climate data quality thresholds as these help frame our reported uncertainties.

-L256-260: Could there be seasonal bias in the gridded means, CVs etc? For example, there were observations in March 2018 and March 2019, but only in November 2017 (not 2018)? Are the spring and summer months overrepresented relative to the fall and winter months, given the data gap between October 2018 and March 2019? Also, I think some explanation of the largest data gaps is warranted- what happened?

Thank you, we agree the gridded means likely over-represent spring and summer relative to the autumn and winter conditions. We have added a statement in Section 3.3.1 to reflect this and a point in the conclusions about the need for more autumn and winter data.

-L278: "three terms in Equation 1…"

Corrected.

-L288-291: these buffer factors should be defined and explained, and the global values provided. This would be a good job for the Supplementary Material.

Thank you. We failed to include values for the salinity and alkalinity buffer factors. We have added these and moved this material to the Supplementary Material.

-Section 2.3.2: I wonder if this whole section could go in the Supplementary. It's important, but does not directly tie into the results presented, and a small subset of this section referenced to the Supplementary might be able to orient the reader to the big picture (thermodynamic vs. biophysical $pCO_2$ drivers).

Thank you, in order to decrease the length of the manuscript, we have followed the reviewer's suggestion and moved this section to the Supplemental Material.

-L310: where did this growth rate come from?

We have clarified that the 2.5 ppm/yr growth rate is the average of annual values from ESRL over the 2014-2019 period.

-L310: or $pCO_2$ growth in coastal zones may not be apparent at all (i.e. Salisbury and Jonsson 2018)

Thank you, we have added this reference.

-L351-353: this sentence can be removed

We have removed this sentence.

-L354: the Alexander Archipelago isn't shown in Figure 1, as far as I can see

We have removed the term Alexander Archipelago and instead use only southeast Alaska throughout the manuscript.

-L393: can you estimate the relative strengths of the seasonal blooms in each year from satellite data?

We thank the reviewer for this comment but believe the additional analysis of satellite data is beyond the scope of our study and rely on the biogeochemical signals described to explain the slight evidence of inter-annual variability in the dataset. We note that our description of the inter-annual variability is limited to the 2018 and 2019 spring and summer, and now highlight more clearly that inter-annual variability needs to be further evaluated with future research.

-Figure 4 caption: these pHT values are at in-situ temperature, correct? In general, I think the figure captions can be expanded to explain the plots more, and perhaps even take some of the explanation out of the text itself

Thank you. Section 3.3.1 explains that $pH_T$ is computed using intake temperature (i.e. SST) and so all discussion of pH in this manuscript is at in-situ/SST conditions. We have carefully reviewed all legends and expanded them to include better explanation of the axes to help orient the reader, as was previously suggested by the reviewer.

-L449: were temperature and salinity the same between years?

Thank you, we have added a statement to Section 4.1 explaining that patterns in temperature and salinity appeared similar between the spring and summer seasons in 2018 and 2019 despite the occurrence of a marine heatwave beginning in late 2019 in the North Pacific basin.

-L465-466: I think this is repeated from earlier.

Section 3.3.2 describing the calculation of thermodynamic and biophysical drivers has been relocated to the Supplemental Material, and the remaining sentences now are no longer repetitive with this statement.

-L548: Serguis Narrows and Wrangall Pass are not on the map in Figure 1

These locations have been added to the map in Figure 1. Note the name Wrangell Pass has been corrected to Wrangell Narrows.

-L590: do previous studies indicate that seater $pCO_2$ in this area is increasing at the rate of the atmospheric increase?

There are no previous studies examining rates of change in seawater pCO2 in this area. This is why we highlight for the reader the difference between "forced" and observed trends. Our ToD estimates are "forced" values based on a prescribed pCO2 increase. Over time, observed rates of change in pCO2 may lead to ToD determinations that differ from the "forced" estimates we provide (see Turk et al 2019). However, we argue that ToD estimates provided here are meant to guide observing efforts and make the case for particular regions being established as monitoring sites because of the short "forced" ToD values.

-L600: "additional change might be anticipated is greenhouse gas emissions are reduced to reach…"

Corrected.

-L626-644: this section had me a little confused. What do the changes in Figure 8 over time indicate?  Different surface water sources?  Direct addition of anthropogenic $CO_2$ via summer $pCO_2$ drawdown vs. longer time scales for mixing/advection?

Thank you for this comment. The top panel of Figure 8 shows how the anthropogenic CO2 content varies along the ferry transit over the time span of the dataset. The lower panel of Figure 8 shows the year when omega_arag < 1 first appeared. This was estimated by first calculating the preindustrial omega values, and then the anthropogenic CO2 content and resulting change in omega_arag for each year over the industrial era. We use this calculation to show that the most weakly-buffered areas have likely exhibited corrosive conditions over the entire industrial era, and that these hot spots are being amplified by continued anthropogenic CO2 addition. We have revised this paragraph to clarify these points.

-L650 "limit"

Corrected.

-L663: the analysis of 2035 acidification levels don't consider temperature increase, correct?

Yes this is correct. Our theoretical calculations only consider changing atmospheric CO2, as stated in the second sentence of Section 4.4.

-L669-671: I wonder if averages are the best indicator here. What about a median with range, or one standard deviation? There is surely some variability that these averages are not capturing.

Thank you, we have adjusted these statements to include the average +/- 1 standard deviation.

-L673: "have been"

Corrected.

-L673: should pH change values be negative?

Corrected.

-L725-727: I think the freshwater alkalinity variability is discussed, but could be highlighted more.

Thank you, we have highlighted that this is an important next step for research in this area, particularly in southeast AK. We have also included a point about considering deviations from the global S-Ca$^{2+}$ relationship following comments from Reviewer 2.

-L728: "dominant"

Corrected.

-L740: large portions of the Inside Passage show omega<1 beyond 2035, right?

This is true, but we limit our theoretical evaluation of 1.5C acidification conditions to 2035, as going beyond 2035 with our current emissions trajectory (SSP5-like scenario) implies even higher atmospheric CO2 achieved by exhausting more than the 1.5C remaining carbon budget.

-This conclusions section summarizes the paper well, but doesn't do much to point the way forward from here. Some of that is done in the Discussion, which could be moved here instead.

Thank you, we have enhanced the discussion of the path forward in the conclusions section to reflect this comment.

Supplementary Figure S2: Could the bottle alk here below salinity 22 be used to refine the Alk(S) relationship, by basically developing a different relationship at S<22?

We thank the reviewer for this comment and feel that it is an excellent next step for this research. For our current analysis, we do not employ this approach because we felt the number of data points is still too limited. But future efforts will work to expand marine CO2 system measurements within the low S waters of southeast Alaska.

Supplementary Figure S11 caption: "therefore"

We have made this correction.

Reviewer 2:

In this paper, Evans et al. presented the full carbon system parameters along the Inside Passage of the Pacific Northwest coast of North America from November 3, 2017 to October 2, 2019. They examined their seasonal and temporal changes and discussed their controlling mechanisms, and estimated the Time of Detection. They even projected the conditions when atmospheric CO2 reaches a level that exhausts the remaining 1.5°C carbon budget. The paper is very well-written and the data will contribute to the understanding of ocean acidification status in an important region.

Major comments:

1. Please consider adding a plot showing the internal consistency of the measurements.

We thank the reviewer for this comment. Internal consistency of the marine CO2 system is evaluated by "over-determining" the system using 3 or more distinct measurements of marine CO2 system parameters. For instance, internal consistency could be evaluated with measurements of pH, pCO2, and TCO2 made on the same seawater sample. During this study, there was never 3 distinct measurements of the marine CO2 system made; the system was never over-determined. During the first year, bottle samples were analyzed for pCO2 and TCO2 and pCO2 was continuously measured using the GO8050 on the ferry. During the second year, pH and pCO2 were both continuously measured on the ferry with no bottle samples for TCO2. Therefore, the comparison we can make requires alkalinity from an empirical relationship, which we have done and show in Supplemental Figures S2 and S3. S2 shows the comparison between alkalinity determined from the pCO2 and TCO2 measurements and alkalinity determined from a regional alkalinity-salinity relationship. S3 shows the comparison between directly measured pH and pH computed using continuous pCO2 measurements and alkalinity determined from a regional alkalinity-salinity relationship.

2. [Ca2+] in the open ocean can be assumed to be conservative with salinity, which is the basis for the CO2SYS calculation. However, in the coastal ocean, especially in low salinity areas, the [Ca2+]-salinity relationship can be quite different from region to region (Dillon et al., 2020). I wonder if the authors could find any directly measured calcium data in the region, so as to improve the uncertainties of the calculated aragonite saturation states.

We thank the reviewer for this comment and agree that deviation for the global salinity/Ca$^{2+}$ relationship is a source of uncertainty, particularly in the low-salinity, glacial melt influenced regions of southeast AK. We see this as an important next step to the work being done in southeast AK, which includes building a building a new regional algorithm for alkalinity that better captures the low salinity water in this region. We have highlighted this point in our discussion and conclusions, and now include the Beckwith et al 2019 reference, which we felt better captured the issue around riverine Ca$^{2+}$ contribution.

3. The estimation of anthropogenic CO2 levels in a coastal setup is kind of surprising. The current method to estimate anthropogenic CO2 is mainly designed for the open ocean. For the coastal ocean, pCO2 level is strongly controlled by many other processes, such as river and ground water input, eutrophication, benthic processes, etc. I'm not even sure if it is a good idea to provide such

an estimate. The same goes true for the estimation of the conditions when atmospheric CO2 reaches a level that exhausts the remaining 1.5°C carbon budget.

We thank the reviewer for expressing concern regarding the estimation of anthropogenic CO2 and theoretical evaluation of the 1.5C acidification level. However, as described in our methods, the approach we use has been applied to coastal datasets (see Takeshita et al., 2015, Pacella et al., 2018, and Evans et al 2019). The reviewer is correct that multiple processes unique to the coastal ocean drive large variability in TCO2 that make evaluation of the anthropogenic CO2 signal challenging. But the approach used accounts for this variance in the delta-TCO2 disequilibria term, and makes the key assumption that this term is constant, meaning the drivers of TCO2 disequilibrium with the contemporary atmosphere are not changing over time. While this assumption may be less valid in certain regions, such as areas that have experienced an increasing eutrophication signal, the assumption appears to hold in our region based on the comparison to other recent estimates of anthropogenic CO2 content from Feely et al 2010 for the Salish Sea (Puget Sound) and from Carter et al 2019 for the coastal North Pacific surface water. The surprise for us were the low values found in areas influenced by glacial melt, which is understandable given these waters have limited ability to acquire an appreciable anthropogenic CO2 signal. We clearly state these results are theoretical and do not account for change in other drivers such as increasing temperature or changing freshwater input. We feel that inclusion of this theoretical analysis is important for evaluating the contemporary anthropogenic CO2 content, describing the degree of acidification in our region and how it varies spatially and seasonally, and for guiding experimental work on appropriate regional targets that should resemble a 1.5C warmed world.

4. Please consider creating a separate section called Study site and move the current information about the site from Introduction to the new section.

Thank you for this comment. Reviewer 1 had a similar comment, and we have addressed this by adding a "Study Region" section to the manuscript.

Minor comment:

1. Throughout the paper, please italicize the "p" within "pCO2".

Thank you for this comment. Italicizing the p would denote it as a mathematical operation, which is not correct in this case. We have seen this appear in the literature and believe it stems mainly from author preference. Because of this reason and the potential confusion with mathematical operation, we maintain "p"$CO_2$ throughout the manuscript.

2. Throughout the paper, please replace "concentration" with "content", if the values are reported as per kg SW. Concentration is a term for per volume based measurements.

We have switched "concentration" for "content".

3. Hydrogen ion concentration -> Total hydrogen ion content (assuming you are talking about the amount estimated based on pH on Total Scale)

We have specified "total hydrogen ion content" on Line 45 and adjusted [H$^+$] to [H$^+$]$_T$ throughout the manuscript.

4. Line 39: Replace "412 ppm in 2019" to "414 ppm in 2020".

We have updated this statement to reflect the 2020 value.

5. Line 40: Please recheck this number. I remember it is more like 600 GtC instead of 700 GtC. I could be wrong though.

Corrected to 690+/-80 GtC as per Friedlingstein et al 2021.

6. Line 41: Replace "Friedlingstein et al., 2020" with "Friedlingstein et al., 2021".

We have updated to the 2021 reference.

7. Line 48: "saturation state" -> "saturation states".

We have made this correction.

8. Line 48: If you choose to use the word "more", you'll better off finding a place to mention calcite?

We have edited this statement and removed the word "more".

9. Line 52: For the change of aragonite saturation state, it is better to report a percentage number. After all, its baseline varies significantly across the global ocean. A change of 0.53 could mean dramatically different things in the polar region compared to the tropical region.

While we agree with the reviewer's point, we retain reporting the absolute decrease in global average aragonite saturation state as this is what is reported in the literature by Lauvset et al., 2020.

10. Line 252: The uncertainty of [H+] needs a unit.

Thank you, we have added this missing unit.

11. Line 258: Please specify the gridding method you used.

The description of the gridding method was adjusted to "observations were gridded by isolating and averaging data within 0.03° by 0.03° grid cells".